# TNF controls a speed-accuracy tradeoff in the cell death decision to restrict viral spread

Jennifer Oyler-Yaniv[1,4], Alon Oyler-Yaniv[1,4], Evan Maltz[1] & Roy Wollman ⓘ [1,2,3✉]

Rapid death of infected cells is an important antiviral strategy. However, fast decisions that are based on limited evidence can be erroneous and cause unnecessary cell death and subsequent tissue damage. How cells optimize their death decision making strategy to maximize both speed and accuracy is unclear. Here, we show that exposure to TNF, which is secreted by macrophages during viral infection, causes cells to change their decision strategy from "slow and accurate" to "fast and error-prone". Mathematical modeling combined with experiments in cell culture and whole organ culture show that the regulation of the cell death decision strategy is critical to prevent HSV-1 spread. These findings demonstrate that immune regulation of cellular cognitive processes dynamically changes a tissues' tolerance for self-damage, which is required to protect against viral spread.

[1] Institute for Quantitative and Computational Biosciences, University of California, Los Angeles, CA, USA. [2] Department of Integrative Biology and Physiology, University of California UCLA, Los Angeles, CA, USA. [3] Department of Chemistry and Biochemistry, University of California UCLA, Los Angeles, CA, USA. [4] These authors contributed equally: Jennifer Oyler-Yaniv, Alon Oyler-Yaniv. ✉email: rwollman@ucla.edu

Rapid detection of viral infection is needed to minimize spread and associated damage. At the same time, evolutionary pressures constantly enhance the arsenal of strategies that viruses use to evade and manipulate host detection[1,2]. The adversarial co-evolution between host detection and viral avoidance creates a challenging and time-dependent classification problem for host cells that likely operate close to the detection limits imposed by the inherent stochasticity of biochemical reactions. Rapid decision making at the noise limit is likely to result in false-positive classification i.e., a decision that a virus is there when it is not, followed by unnecessary cell death and bystander tissue damage. How cells navigate the tradeoff between decision speed and accuracy to achieve an optimal classification strategy has not been explored.

The initial detection of viral invasion in a tissue is accomplished by macrophages and other innate immune sentinels[3]. These specialized cells secrete cytokines and other inflammatory mediators to alert nearby cells of invading pathogens. Alerted cells then execute diverse strategies to prevent viral spread including restriction of viral entry, direct inhibition of viral gene expression, prevention of viral genome replication, prevention of viral egress, and death of infected cells[4,5]. These defense mechanisms are activated in an inducible manner as they have the potential to interfere with normal host cell physiology[6–9]. We asked how cells balance the competing goals of antiviral activity and maintenance of normal physiology in response to macrophage-derived inflammatory cytokines. We focused on Tumor Necrosis Factor α (TNF), because it is an important antiviral cytokine[10], yet can also cause significant damage to healthy host tissues due to its role regulating cell death pathways[11]. This relationship is exemplified by the observation that TNF-blocking drugs have shown remarkable success for treatment of autoimmune disorders, yet also increase susceptibility to certain pathogens[12]. Further, excessive production of TNF has been implicated as an important driver of cell death and tissue destruction during infection with hyper-virulent virus strains such as the H5N1 and 1918 pandemic influenza subtypes[6–8].

Here, we combine experiments in cell culture with mathematical modeling and live, whole-organ light-sheet microscopy to examine the regulation of the cellular decision to undergo cell death during viral infection. We show that TNF regulates a tradeoff between the cell death decision speed and accuracy, such that infected cells die faster at the expense of increased death of uninfected bystander cells. Importantly, the regulation of this speed-accuracy tradeoff in single cells is critical to limit viral spread throughout a whole population. Our work demonstrates that the immune system regulates the execution of individual cellular decisions to tune a tissues' tolerance for self-damage based on pathogen threat.

## Results

**TNF secretion by macrophages limits viral spread.** We first quantified the effect of macrophage density, inflammatory status, and TNF secretion on viral spread throughout a population of cells. We co-cultured fibroblasts with either naive or activated mouse bone marrow-derived macrophages (BMDM) at different densities and infected them with a low multiplicity of infection (MOI 1) of Herpes Simplex Virus-1 (HSV-1). We used HSV-1 recombinant viruses expressing capsid protein VP26-fluorescent protein fusions to monitor viral infection using time-lapse microscopy[13]. As expected, activated, but not naive, BMDM restricted viral spread (Fig. 1A, B, Supplementary movies 1–4).

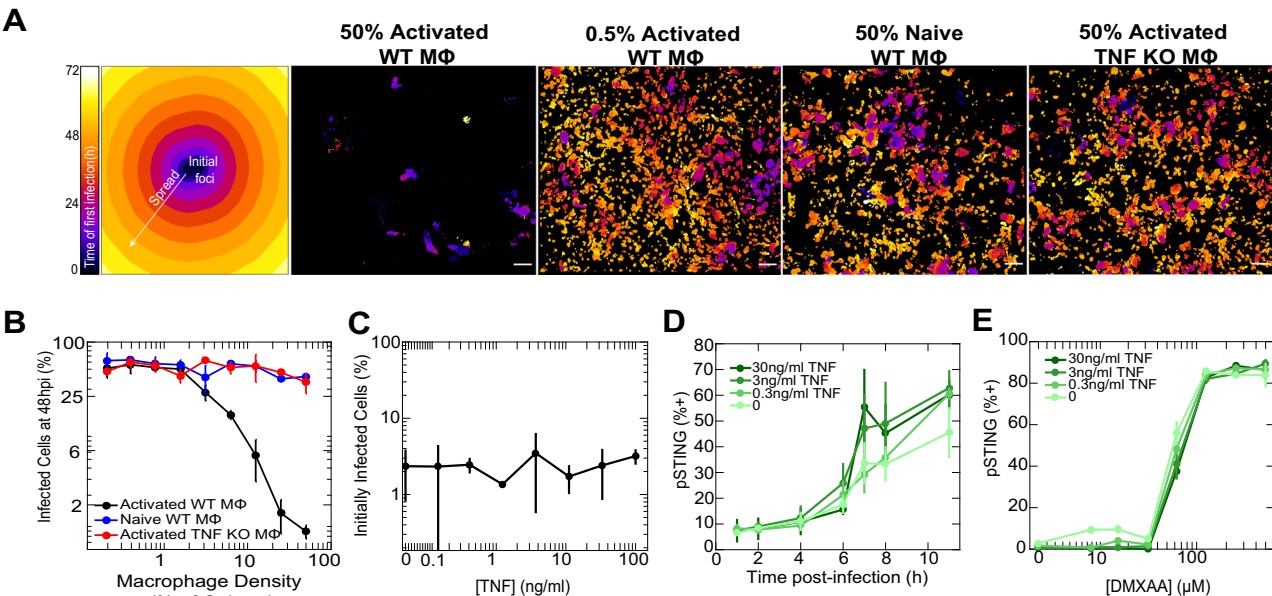

**Fig. 1 Activated macrophages restrict viral spread by production of TNF. A** Representative temporal color-coded maps of viral infection time for selected BMDM-fibroblast co-culture conditions (scale bar, 100 μm). The key (far left) demonstrates the color-coding scheme to interpret the remainder of **A**. Regions infected early during infection are colored darker whereas regions infected later during the experiment are colored lighter. Black regions represent areas without infection. **B** Quantification of viral spread across all BMDM-fibroblast co-culture conditions. Infection was determined based on fluorescence intensity of tagged viral VP26 in fibroblasts at 48hpi. **C** Quantification of cells initially infected only from virus (MOI=1) added to the media after being pulsed overnight with different doses of TNF. Infection was determined based on fluorescence intensity of tagged viral VP26 in fibroblasts at 24hpi. **D** Quantification of cells that stained positive for active, phosphorylated STING (pSTING) after being pulsed overnight with different doses of TNF and infected with MOI 10 of HSV-1. pSTING+ cells were identified based on antibody fluorescence intensity at the specified timepoint post-infection. **E** Quantification of cells that stained positive for pSTING after being pulsed overnight with different doses of TNF and exposed to dose titrations of the pSTING agonist DMXAA. pSTING+ cells were identified based on antibody fluorescence intensity 1.5 h after DMXAA treatment. Data in **B**–**E** are mean ± s.d.

Activated macrophages produced high levels of TNF (Supplementary Fig. 1A), suggesting that this cytokine is a potential key regulator. Indeed, BMDMs from mice that lack TNF, and WT BMDMs supplemented with TNF neutralizing antibodies did not restrict viral spread (Fig. 1A, B, Supplementary Fig. 1B). These results show that the restriction of viral spread by activated, inflammatory macrophages depends on the production of TNF. Interestingly, the protective effect of TNF against viral spread was not mediated by reduced cell infectivity or by enhanced sensing of either HSV-1 or synthetic viral ligands (Fig. 1C–E, Supplementary Fig. 1C–E).

**TNF transitions cells into a primed-to-death cell state.** TNF has a well-established role regulating cell death pathways (Supplementary Fig. 2A–D)[11]. Therefore, we investigated whether viral defense is related to TNF increasing the cellular propensity to die. TNF simultaneously activates opposing pro-death signaling and a pro-survival transcriptional response[14–18]. Consistent with previous results, exposure to a saturating dose of TNF (100 ng/ml) for 6 h killed only 45% of the cells, yet co-treatment with TNF and Actinomycin D, to block transcription, killed nearly all cells in the population (Fig. 2A)[11]. This suggests that TNF activates death pathway signaling in most of the cells, but the execution of death is restrained by cellular production of pro-survival proteins. What biological function could result from the simultaneous activation of antagonistic pathways? We reasoned that the primary function of TNF may not be to kill cells per se, but to shift cells into a "primed" state, conferring an increased propensity to die upon exposure to additional death signals. To test this

hypothesis, we co-treated cells with a concentration of TNF too low to cause any cell death (0.3 ng/ml), and the Inhibitor of Apoptosis Protein (IAP) antagonist LCL-161[19] IAPs are key restriction points for TNF-mediated cell death:[20–23] cIAP1 and cIAP2 restrict RIPK1 participation in the death inducing signaling complex[24–26], and XIAP restricts Caspase 3 activation[27]. This experiment revealed that although present at too low of a concentration to kill cells alone, TNF sensitizes cells to be killed by LCL-161 (Fig. 2B). Importantly, TNF is necessary for LCL-161 to exert any cytotoxic effect as cells not treated with TNF are insensitive to LCL-161. While the use of LCL-161 recapitulates the qualitative results obtained by Actinomycin D, contributions from other mechanisms, such as regulation of caspase-8 by FLIP were not excluded and could be involved in the creation of 'primed to death' cell state.

We next asked if the priming effect of TNF depends on ligand exposure or represents an altered cell state that does not rely on continued receptor ligation. To test this, we treated cells for 24 h with the same low dose of TNF (0.3 ng/ml), then washed cells and rested them overnight (~12 h) before treating with LCL-161 (Fig. 2C). Importantly, treating 3T3 fibroblasts with TNF does not cause cells to produce it[28]. Even after a rest period of 12 h, TNF-treated cells retained sensitivity to LCL-161, indicating that constant exposure to the ligand is not necessary for sensitization to death (Fig. 2D). Consistent with this result, cultures treated with higher doses of TNF exhibited Caspase-8 cleavage and cell death that persisted for 24–36 h after the cytokine was washed out (Fig. 2E, F). Finally, we quantified how long it takes for TNF-primed cells to lose this sensitized state. Cells were treated for 24 h with 1 ng/ml TNF and then washed (Fig. 2G). Actinomycin

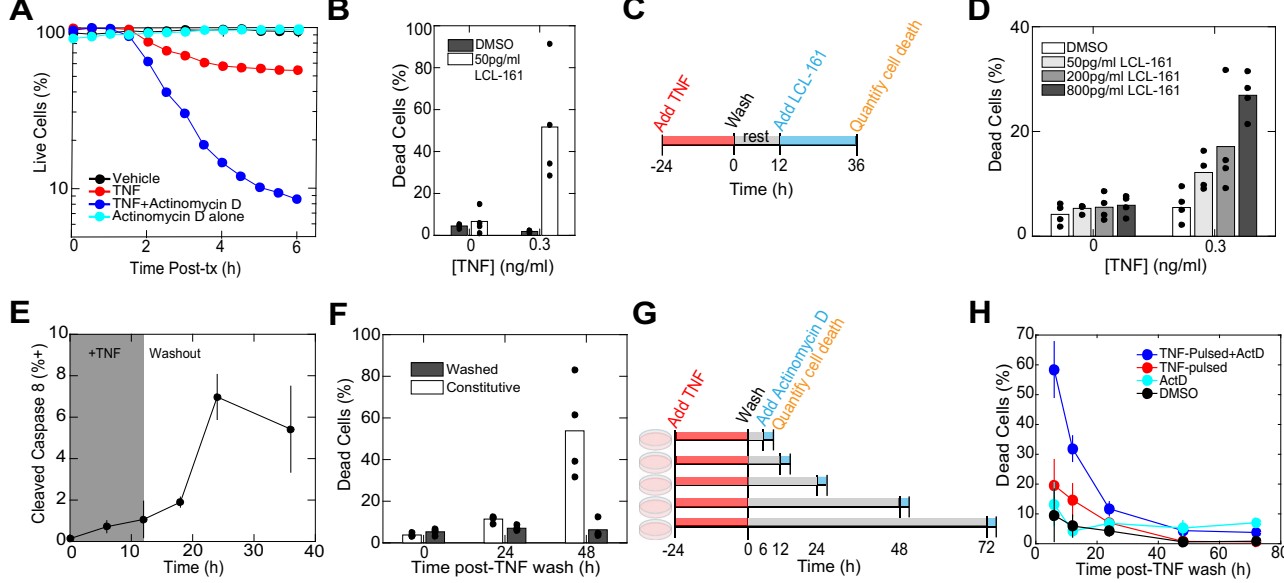

**Fig. 2 TNF transitions cells into a primed-to-death cell state. A** Quantification of cell death in NIH 3T3 fibroblasts after treatment with 100 ng/ml TNF or TNF supplemented with 0.25 μg/ml Actinomycin D. Cell death was quantified over time based on Hoechst 33342 fluorescence intensity which increases sharply on nuclear condensation. **B** Quantification of cell death in fibroblasts 24 h after simultaneous treatment with 0.3 ng/ml TNF and 50 pg/ml LCL-161. Cell death was quantified based on incorporation of a nucleic acid stain (sytox green). **C** Diagram of experiment for data shown in **D**. 3T3 fibroblasts were treated for 24 h with 0.3 ng/ml TNF and then washed and rested overnight. The next day, LCL-161 was added at the indicated doses and cell death was quantified after 24 h of LCL-161 exposure. **D** Quantification of cell death for experiment described in **C**. Cell death was quantified based on incorporation of sytox green. **E** Quantification of 3T3 fibroblasts that stained positive for cleaved Caspase-8 during and after treatment with 100 ng/ml TNF. The shaded region indicates the time period before TNF was washed out. Cleaved Caspase-8 was detected by immunostaining and cells were distinguished as positive by setting a threshold on the antibody fluorescence intensity. **F** Quantification of cell death for 3T3 fibroblasts treated constitutively with 30 ng/ml TNF or washed after overnight treatment with 30 ng/ml TNF. Cell death was quantified based on incorporation of sytox green. **G** Diagram of experiment for data shown in **H**. 3T3 fibroblasts were treated with 1 ng/ml TNF for 24 h and then washed. At different times after washing, 0.25 μg/ml Actinomycin D was added to cells for 4 h before quantifying cell death. **H** Quantification of cell death for experiment described in **G**. Cell death was quantified based on incorporation of sytox green. **B**, **D**, **F** bars denote the mean of replicates (filled black circles). **E**, **H** are mean ± s.d.

D was added to cells at different timepoints post-wash and cell death was quantified after 4 h. TNF-treated cells retained sensitivity to Actinomycin-mediated death for hours after the ligand was removed (Fig. 2H). This demonstrates that the death-sensitized cell state persists for about 24 h after removal of TNF. Collectively, these data illustrate that TNF transitions cells into a reversible, ligand-independent "primed-to-death" cell state in which pro-death pathway activity is counteracted by cellular production of pro-survival factors.

**TNF regulates a tradeoff between decision speed and accuracy.** Rapid cell death following viral detection is one way to prevent completion of the viral life cycle[5,29] and therefore we hypothesized that the TNF-induced "primed for death" state allows cells to accelerate their commitment to death in certain contexts. To

measure the speed of cell decision, we treated fibroblasts with dose titrations of TNF, infected them with MOI 10 of HSV-1, and imaged viral infection and cell death for 48 h. MOI 10 causes a synchronous and uniform infection minimizing the impact of viral spread, which simplifies interpretation. We tracked individual cells and identified the timing of infection and death (Fig. 3A). We observed that TNF treatment shortened the time interval between infection and death. TNF did not shorten the time from the increase in caspase-8 activity to death, which was much shorter than the time from infection to caspase-8 activation[21] (Supplementary Fig. 3A–C). We will refer to the time interval between infection and death as the death decision latency. To further establish that the decision latency is a rate-limiting step we fitted the measured single-cell decision latencies to an exponential distribution and found that the average latency matched the rate of population decline (Fig. 3B, C,

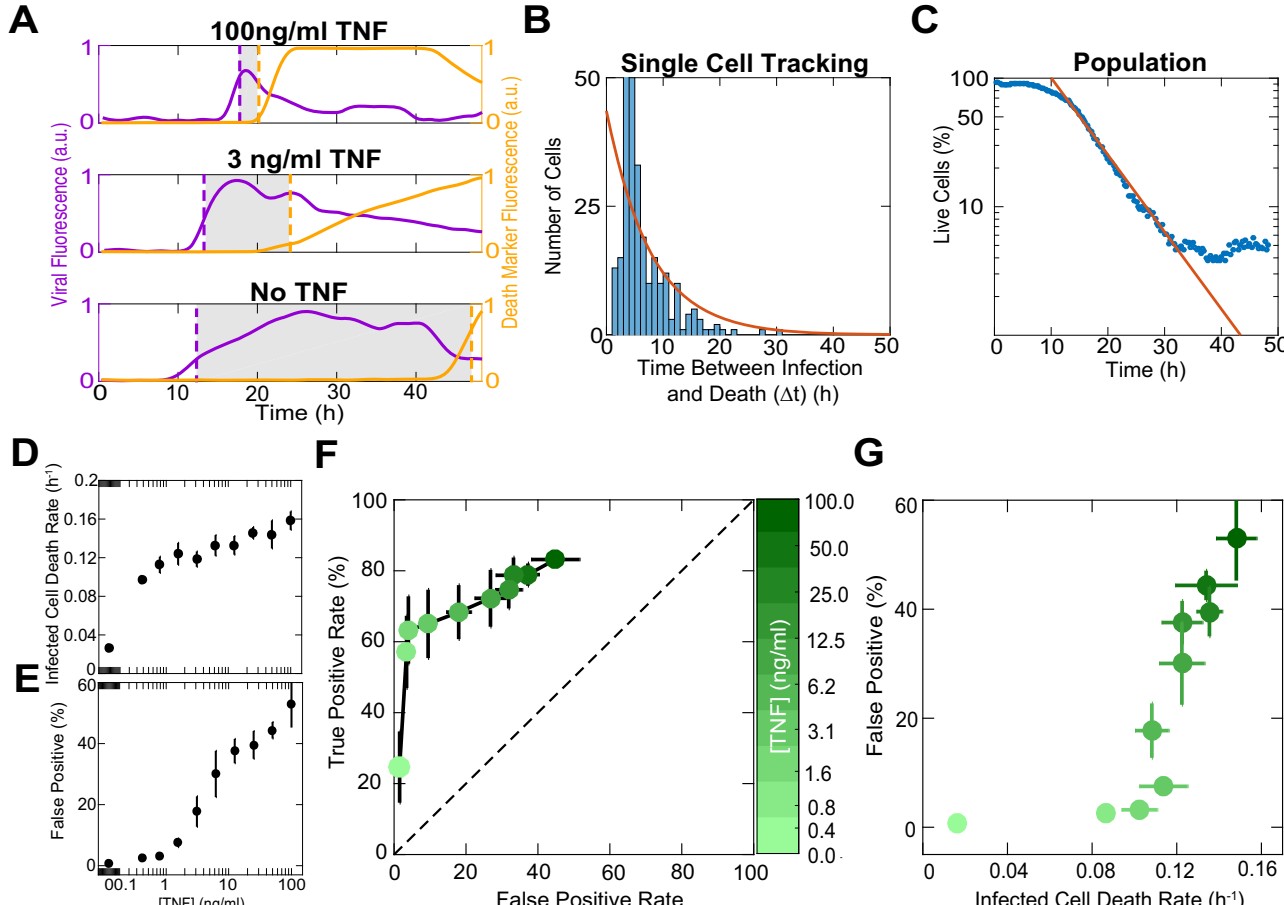

**Fig. 3 TNF regulates the tradeoff between death decision speed and accuracy. A** Representative single-cell traces of 3T3 fibroblasts treated with different doses of TNF and infected with MOI 10 of HSV-1. Purple and orange lines represent the fluorescence over time of the virus and nucleic acid stain (sytox green, death marker), respectively. Dotted purple and yellow lines indicate the timing of initial infection and death. The timing of initial infection was determined by setting a threshold on viral VP26 fluorescence intensity. The timing of death was determined by setting a threshold based on incorporation of nucleic acid stain (sytox green) and nuclear condensation identified as a sharp increase in Hoechst 33342 intensity (for details on thresholding, see methods). Shaded regions represent the time interval between infection and death (death decision time, Δt). **B** Distribution of death decision times obtained by single-cell tracking of 3T3 fibroblasts treated with a single dose of TNF (50 ng/ml) and infected with MOI 10 of HSV-1. Orange line shows an exponential fit of the data. **C** The same sample that generated **B**, plotted as the percentage of live 3T3 fibroblasts over time. Orange line shows the same exponential fit plotted in **B**. **D** Death rates obtained from exponential fits of 3T3 fibroblasts treated with dose titrations of TNF and infected with MOI 10 of HSV-1. **E** Quantification of the percentage of uninfected 3T3 fibroblasts killed by TNF (false positives) at 24 h post treatment. Cell death was determined by incorporation of a nucleic acid stain (Sytox green or red), accompanied with nuclear condensation measured as a sharp increase in Hoechst 33342 intensity. **F** Receiver Operating Characteristic (ROC) curve where false-positive rate represents 3T3 fibroblasts killed by TNF at 24 h post treatment and true positive rate represents infected cells that die during the first viral life cycle (10 h after initial infection is called based on viral VP26 fluorescence intensity). **G** Speed versus accuracy tradeoff shown by plotting the percentage of false-positive 3T3 fibroblasts (from **E**), versus the death rate for infected 3T3 fibroblasts (from **D**). **D**–**G** are mean ± s.d. **F**,**G** colorbar shows TNF concentration.

Supplementary Fig. 3D, E, G). From these population measurements, we concluded that TNF dose-dependently shortens the death decision latency (Fig. 3D).

Our data show that TNF dramatically shortens the decision latency specifically following viral infection (Fig. 3D, Supplementary Fig. 3D, E, J). We asked if there are downsides to this short-latency state that constrain the speed of cellular decision making. A tradeoff between decision speed and accuracy is well established in other cognitive systems[30,31]. We analyzed whether the increasing speed in execution of cell death comes at a cost of accuracy. Two possible decision errors are false negative—infected cells that do not die—or false positive—cells that die without being infected. In our MOI 10 experiments, there were almost no infected cells that escaped death (false negatives), by 48 h post-infection (hpi) (Supplementary Fig. 3E). On the other hand, the rate of false-positive errors increased in a TNF dose-dependent manner (Fig. 3E, Supplementary Fig. 3F). Using a receiver operator curve (ROC), a standard assessment of a classifier performance, we saw a tradeoff between accurate decisions (true positives) and erroneous ones (false positives) (Fig. 3F). Contrasting the increase in error rates with the increase in decision speed reveals that the cellular decision to die is subject to a tradeoff between speed and accuracy (Fig. 3G). Our data therefore demonstrate that cell fate decisions are not only subject to the speed-accuracy tradeoff but that this tradeoff is-also tunable and regulated.

**Regulation of the speed-accuracy tradeoff prevents viral spread.** We next investigated how the dynamics of viral spread are affected by TNF modulation of the death decision strategy. We confirmed that TNF-mediated restriction of viral spread depends on cell death (Supplementary Fig. 4A, B), supporting that viral spread depends on the interplay between infection and death. To better understand this connection we used a stochastic and spatially explicit mathematical model of viral spread (Fig. 4, Supplementary Table 1, Supplementary Note 1). In our model, each cell can be in one of four states: healthy, infected, dead following infection, or dead but uninfected (Fig. 4A). The cells are organized on a uniform hexagonal grid and the simulation is initialized by infecting the central cell on the grid. These conditions are analogous to a low MOI infection where a small fraction of cells form an initial seed from which the virus spreads radially in a juxtacrine manner. The viral load of each infected cell grows linearly and deterministically while the cell is alive (Supplementary Fig. 4C). Healthy cells can be infected by their infected neighbors and the probability of infection is determined by the total viral load of a target cell's live neighbors (Supplementary

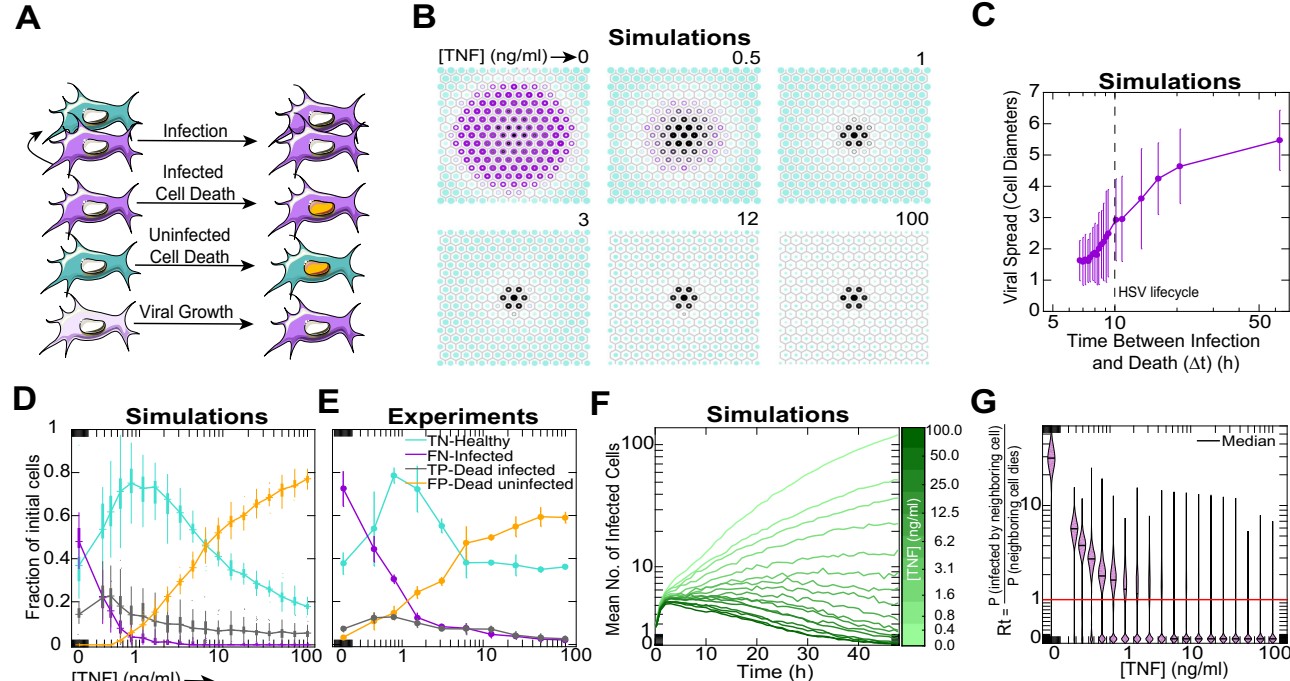

**Fig. 4 Modulation of the tradeoff between cellular death decision speed and accuracy controls viral spread throughout a population. A** Diagram of the reactions in our spatial stochastic model. All model parameters were measured in independent experiments shown in Supplementary Fig. 4C-F, J, summarized in Supplementary Table 1. **B** Ensemble simulation results for different TNF concentrations. Purple area and intensity correspond to the probability of the cell being infected. Black area and intensity correspond to the probability of the cell being dead following infection. Cyan area and intensity correspond to the probability of the cell being alive and healthy. **C** Quantification of viral spread as a function of TNF concentration, mean ± s.d. **D** Boxplot summary of simulation results for a dose titration TNF conditions showing the percentages of healthy, infected, dead following infection, and dead but uninfected cells. Central mark denotes the median, boxes denote the upper and lower quartiles, whiskers extend to the most extreme points not considered outliers (≥1.5 interquartile range from top/bottom of box), and outliers are plotted as points. **E** Quantification of the percentages of healthy, infected, dead following infection, and dead but uninfected 3T3 fibroblasts treated with dose titrations of TNF and infected with MOI 1 of HSV-1. Data are mean ± s.d. **F** Simulation of the dynamics of actively infected cells for different concentrations of TNF. Lines represent the mean values of 500 simulations. **G** Effective reproductive rates (Rt) distributions in different conditions represented as violin plots, which were generated with non-negative support and a bandwidth of 0.05. In this context, Rt represents the ratio of the probability that a cell is infected by its infected neighbor to the probability of that neighboring cell dying. The red line denotes a separatrix: Above it, the virus is more likely to spread further and below it, the virus is more likely to be extinguished. Black lines indicate median. For simulations (**B–D**, **F**, **G**), each condition was simulated 500 times.

**Table 1 Model parameters.**

| Parameter | Meaning | Source |
|---|---|---|
| VI | Viral infectivity | Supplementary Fig. 4D, Supplementary Table 1 |
| VGR | Viral growth rate | Supplementary Fig. 4C, Supplementary Table 1 |
| $\beta_i$(TNF) | Infected cell death rate | Supplementary Fig. 4E, Supplementary Table 1 |
| $\beta_b$(TNF) | Uninfected cell death rate | Supplementary Fig. 4E, Supplementary Table 1 |

Fig. 4D, Supplementary Table 1). The death rates of individual cells depend on their infection status, their exposure to TNF, and are directly interpolated from measured values (Supplementary Fig. 4E, F, Supplementary Table 1). All model parameters are based on individual cell decisions and were independently calibrated in separate experiments (Table 1, Supplementary Fig. 4C–F, Supplementary Table 1). Variability in the outcome of individual simulations stems from the stochastic nature of the model.

Our model demonstrates that the TNF-mediated manipulation of death rates rates on a single-cell level is sufficient to explain the emergent arrest of viral spread (Figs. 1A, B, 4B, C, F, G). Our results show a sharp system-wide transition from a virus-permeable to a virus-resistant state, which occurs when the single-cell death decision time dips below the viral life cycle[32] (Fig. 4C). The model also captures a global manifestation of the speed-accuracy tradeoff: it predicts an optimal non-zero TNF concentration that optimizes the balance of costs and benefits (Fig. 4D). This prediction seems counterintuitive as TNF increases the death rate of both infected and uninfected cells, yet at certain concentrations it maximizes the fraction of healthy cells. By rapid killing of infected cells, TNF protects bystander cells from infection at the acceptable cost of a slightly enhanced bystander death rate. To test this prediction, we infected fibroblasts with a low MOI of HSV-1 in the presence of different doses of TNF and observed such an optimal concentration and a striking agreement between the predicted and the observed results ($R^2 = 0.69$, Fig. 4E, Supplementary Fig. 4H). We find it remarkable that a simple model with no free parameters matched independently measured data to such a high degree.

The model demonstrates that this strategy not only has the power to slow or arrest viral spread, but to extinguish it from the entire population of cells altogether. Simulations of actively infected cells over time demonstrate that at higher concentrations of TNF, there are zero infected cells remaining by about 48hpi (Fig. 4F, Supplementary Fig. 4G). We used our model to compute the viral effective reproductive rate (Rt) as a function of different doses of TNF (Fig. 4G). In this context, the effective reproductive rate represents the ratio of the probability that a cell is infected by its neighbor to the probability of that neighboring cell dying. TNF dose-dependently reduces the viral reproductive rate. However, more importantly, this plot demonstrates a divergence in population-wide phenotype based on the TNF concentration. Above the separatrix (Rt = 1, red line in 4 G), the virus will spread, below, it will be completely extinguished from the population. Taken together, our model quantitatively captures how TNF modulation of the death decision in single cells translates to the collective phenotype of viral spread versus extinction.

Finally, we used the model to explore how TNF impacts the infection dynamics during infection by a more- or less-infectious form of the virus. (Supplementary Fig. 4I). To this end, we ran the model with varied viral infectivity rates ranging from 0.05 $h^{-1}$ to 48 $h^{-1}$. At very low viral infectivity rates, the maximal cell survival occurs in the absence of TNF. In this regime, the spread of the virus is so slow that any amount of bystander damage will

cause more harm than the virus itself. As the rate of viral infectivity increases, the optimal TNF concentration also increases. Concurrently the prominence of this optimal concentration decreases as more damage is caused both by the faster spreading virus and by TNFs effect on bystander cells. Finally, beyond a certain level of viral infectivity no amount of TNF can rescue the population, which gets annihilated.

**Prevention of viral spread in the cornea.** We next sought to establish that TNF regulation of cellular decision strategies is sufficient to alter viral spread in a whole live organ. To this end, we developed an ex vivo whole corneal culture model of HSV-1 infection coupled with light-sheet microscopy. In our model, whole corneas are dissected from R26-H2B-mCherry mice and infected with fluorescent HSV-1 (Fig. 5A). The cornea is an ideal model for our experiments: it is transparent, avascular, and is very robust in organ culture conditions. These features enable live imaging of viral spread as tissue clearing methods are not required. Biologically, it is a natural target of HSV-1[33], and key to these experiments, it has minimal innate immune activity once dissected from its source of recruited immune cells[34]. In our experiment we took advantage of this fact and compared corneal response with and without the addition of TNF to the organ culture media, mimicking an impact of recruited inflammatory macrophages[35]. Using a custom-built light-sheet microscope (Supplementary Fig. 5), we imaged viral spread and cell death continuously over 48 h. Time-lapse imaging revealed that HSV-1 spread radially over time and created a pronounced viral nodule containing an inner core of dead cells (Supplementary movie 5, Fig. 5B, C). Viral spread was dramatically reduced in corneas treated with exogenous TNF compared to untreated controls (Supplementary movie 6, Fig. 5B–D). In addition, by 48hpi, almost all virus-infected cells were dead in the TNF-treated conditions compared to only about 50% in untreated corneas, demonstrating that viral spread outpaced cell death (Fig. 5C, E). Using dissociated freshly-isolated corneal epithelia, we validated that the increase in cell death among infected cells stems from a TNF induced increase in decision speed (Fig. 5H), in agreement to our observations in fibroblasts (Fig. 3G). Finally, and similar to our in vitro observation, TNF without infection increased the amount of bystander cell death in the cornea, fully recapitulating the death decision speed vs accuracy tradeoff (Fig. 5F, G). Taken together, these data demonstrate that TNF modulates the death decision strategy of corneal epithelial cells to restrict viral spread in vivo.

## Discussion

The data presented here demonstrate that information provided to cells through macrophage secreted cytokines can alter their classification strategy by dynamically shifting their error tolerance in favor of faster decisions. Importantly, these changes to individual cells' decision processes are required to prevent viral spread in the population.

We only observed a TNF-induced change in cellular decision strategy during viral infection and not in response to other

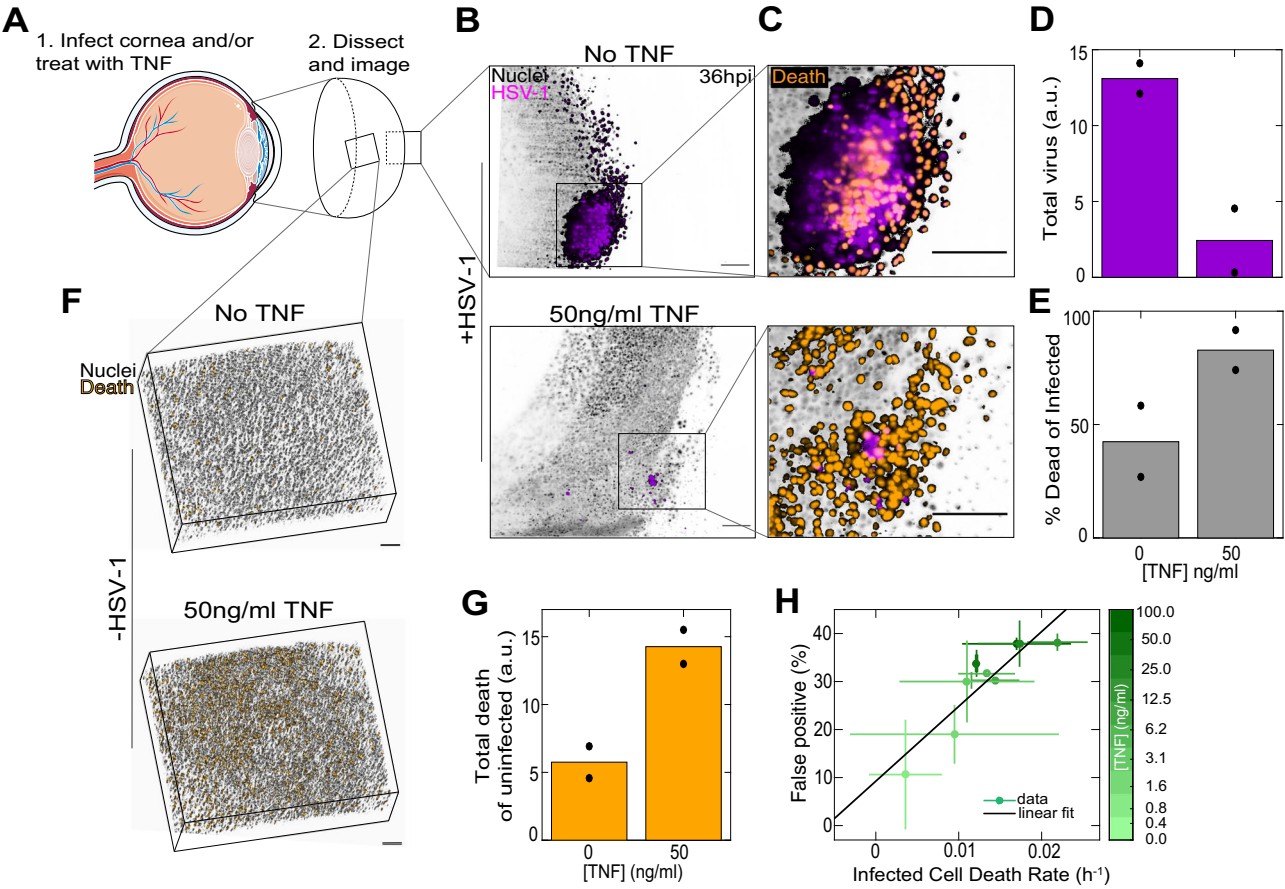

**Fig. 5 TNF alters the cellular death decision-making strategy in the corneal epithelium, which restricts viral spread. A** Cartoon diagram with cornea emphasized to show different perspectives of images. **B** Representative transverse image of R26-H2B-mCherry mouse corneas 36 h after infection with HSV-1 and treated with either 0 or 50 ng/ml TNF. **C** Enlarged regions within images shown in **B** with cell death channel included. **D** Quantification of the total sum of virus pixels for C57Bl/6 corneas treated with 0 or 50 ng/ml TNF, infected with HSV-1, and imaged at 48hpi. **E** Percentage of total virus pixels also positive for Sytox Green death marker. **F** Representative epithelial projections of C57Bl/6 corneas treated with 0 or 50 ng/ml TNF and imaged 24 h after treatment. **G** Quantification of the total Sytox Green intensity for uninfected corneas. **H** Speed versus accuracy tradeoff shown by plotting the percentage of false-positive primary corneal epithelial cells versus the death rate for infected primary corneal epithelial cells. Colorbar represents TNF concentration. Data are mean ± s.d. **D**, **E**, **G** bars denote the mean of replicate eyes (filled black circles). All scale bars, 100 μm.

stresses such as bacterial lipopolysaccharide or DNA damage (Supplementary Fig. 3H, I). One interpretation of this context specificity stems from the fact that viruses are obligate intracellular parasites and death of the host cell halts viral replication. The context specific manner by which TNF acts indicates how the regulation of cellular decision strategies has evolved to provide specific physiological benefits. In our in vitro system, TNF had a significant antiviral effect; however, in vivo, it acts together with other cytokines to restrict viral spread. The relative importance of TNF versus other other cytokines likely varies between different types and stages of viral infection and merits further investigation using in vivo experimental models.

The importance of TNF as an antiviral cytokine is demonstrated by the array of viral adaptations that have evolved to counter its activity[36]. Further, the antiviral effect of TNF has been demonstrated in various animal models of viral infection[37–41]. Nevertheless, in vivo, the antiviral response is executed by the concerted activity of many different cytokines. Type I Interferons are important antiviral cytokines, which exert their effect without killing cells. However, some viruses, such as HSV-1, encode strategies to block its effect[42] and other viruses, such as hepatitis B virus, block its induction altogether[43]. These viral adaptations to Interferons underscore the importance of the immune system to counter infection with many different strategies. To summarize, viruses encode diverse mechanisms to evade the immune response therefore there must be multiple shots on goal to clear pathogens.

In the context of an infection in vivo, spatial effects on cytokine spread are likely to alter some of the effects we observed in vitro. The average concentrations of TNF quantified in blood or serum during viral infection are on the lower end of the range we used in our in vitro experiments[44–48]. However, concentrations of TNF in local cell–cell communications are likely to reach high concentrations as has been inferred for other cytokines in vivo[49], yet tissues are much more heterogeneous because only cells in close proximity to cytokine producing cells are subject to cytokine exposure. We speculate that in vivo, spatial constraints on TNF spread likely provides a layer of protection from TNF-mediated death of healthy bystander cells. However, in disease states where positive feedback creates widespread TNF secretion[28], the entire tissue could be subject to TNFs cytotoxic effects.

Cellular signaling networks are often regarded as information processing units[50–54]. Our work shows that cell fate decisions are subject to the same fundamental principle, the speed-accuracy tradeoff, that governs many other complex decision-making systems[55] including social insects[56,57], humans[30,31], and artificial neural networks[58]. The applicability of such a universal principle to cellular signaling provides further evidence that

these networks are indeed bona-fide information processing systems.

Our data argues that infected cells undergo apoptosis, rather than other forms of cell death. Briefly, cells exhibit active Caspases 8 (Supplementary Figs. 2B, C, 3A–C) and 3 (Supplementary Fig. 2A), and externalize phosphatidylserine (Supplementary Fig. 2D). However, we did not explicitly rule out other relevant forms of cell death, such as necroptosis[59], necrosis[60], or pyroptosis[61]. The key takeaway from our paper is that TNF modulates the speed of cell death during viral infection to restrict viral spread. For this conclusion to be valid, the form of cell death is somewhat inconsequential. One key point is that the dynamics we observe excludes the possibility of TNF and HSV-1 acting orthogonally to independently induce cell death. If that were the case, the rate of cell death in the presence of both TNF and HSV-1 will be the sum of the rates[62] for TNF-treated cells (without infection) and for HSV-1 infected cells (without TNF) which is not the case (Supplementary Fig. 3J). TNF and HSV-1 have a synergistic, rather than orthogonal effect on the death rate. The exact molecular mechanism underlying this synergy merits further investigation but has intriguing therapeutic potential.

Tunability in the death decision strategy can have two potential effects. First, it allows cells to respond dynamically to pathogen threat. In the work presented here, we did not vary the inflammatory signal over time. However, in the context of infection, macrophages dynamically adapt to pathogen load by altering how much TNF is produced[63]. Tunability in the decision strategy will allow cells to dynamically respond to such changes, adding another layer of feedback control. Second, different tissues likely have variable degrees of pathogen exposure and intrinsic tolerance to damage. The tunability of cellular decision strategies allows adapting cellular decision to specific tissue context. Extending the connection between immune regulation of individual cell decision-making strategies and organ level phenotypes to other tissues would pave the way to a more unified theory of tissue immunology.

## Methods

**Resource availability**. Further information and requests for resources and reagents should be directed to and will be fulfilled by the corresponding author, Roy Wollman (rwollman@ucla.edu).

**Mice**. TNF$^{+/+}$-RelA-Venus and TNF$^{-/-}$-RelA-Venus mice (C57Bl/6 background) were provided by Dr. Alexander Hoffmann (UCLA) and used as a source of BMDM. Rosa26-H2B-mCherry hemizygous mouse embryos were purchased from the Riken Laboratory of Animal Resource Development and Genetic Engineering, rederived at the University of California, Irvine Transgenic Mouse Core Facility, and maintained in the UCLA animal facility. C57Bl/6 J mice were purchased from the Jackson Labs. Both male and female mice aged 8–15 weeks were used in experiments. All animal experiments were approved by the UCLA Animal Research Committee and mice were maintained in group housing at an AALAC accredited animal facility in standard SPF conditions.

**Cell culture**. BMDM were differentiated from bone marrow incubated in DMEM with 10% fetal calf serum (FCS), 2 mM L-glutamine, 10U/ml Penicillin, 10 μg/ml Streptomycin, and 10 ng/ml M-CSF (GenScript) for 7 days.

NIH 3T3 cells (mouse, male, ATCC CRL-1658) were maintained in DMEM supplemented with 10% Newborn Calf Serum (NBCS), 2 mM L-glutamine, 10U/ml Penicillin, and 10 μg/ml Streptomycin.

Primary corneal epithelial cells were collected from C57Bl/6 mice by incubating eyes overnight at 4 °C in a 1:1 mixture of DMEM:F12 supplemented with 4 mg/ml Dispase I (Sigma–Aldrich), 10U/ml Penicillin, 10 μg/ml Streptomycin, and 0.025 μg/ml Amphotericin B. Epithelial sheets were peeled from eyes and dissociated in TrypLE for 10 min at 37 °C with gentle agitation. Cells were passed through a 0.45 μm strainer (Corning), washed, and cultured on collagen and fibronectin-coated plates in KSFM supplemented with 5 ng/ml hEGF, 50 μg/ml Bovine Pituitary Extract, and 100 μg/ml Cholera Toxin.

All cells were maintained at 37 °C with 5% CO$_2$. Adherent cells were detached for subculture and experiments using TrypLE. Unless otherwise noted, all cell culture materials were purchased from Gibco/Thermo Fisher.

**Cell stimulation and viral infection**. $12.5 \times 10^4$ 3T3 cells were seeded per well of black 96-well plates in imaging media (Fluorobrite with 10% NBCS, 2 mM L-glutamine, 10U/ml Penicillin and 10 μg/ml Streptomycin) supplemented, where indicated, with dose titrations of recombinant mouse TNFα (Cell Signaling Technologies), 30 μM Z-VAD(OMe)-FMK (Cayman Chemical), 30 μM Necrostatin-1 (Sigma–Aldrich), 0.25 μg/ml Actinomycin D (Thermo Fisher), or the specified doses of LCL-161 (Cayman Chemical). For live imaging, cells were labeled with 20 ng/ml Hoechst 33342 (Thermo Fisher) and death was tracked by incorporation of either 1:10,000 Sytox Green (Thermo Fisher) or 1:5000 Cytotox Green (Essen Biosciences) or Red. Where indicated, cells were infected at the relevant MOI with Herpes Simplex Virus-1 strain KOS bearing either mCherry, tdTomato, or mCerulean-tagged VP26 small capsid proteins (Provided by Dr. Prashant Desai, Johns Hopkins University). DMSO was added as a vehicle control where necessary.

To determine the rate of viral spread from neighbors, one batch of fibroblasts were infected overnight with MOI 6 HSV-1. Infected cells were washed, labeled with 1 μM Cell Trace Far Red (Thermo Fisher) and co-cultured at low density with uninfected cells keeping the total number of cells at $1.6 \times 10^4$ per well.

For TNFα pretreatment experiments, cells were cultured overnight with dose titrations of TNFα, then washed 3× with PBS, and simulated with dose titrations of the synthetic STING agonist DMXAA (Invivogen) for 1.5 h, or infected with MOI 1, 2, or 10 of HSV-1.

BMDM were activated by priming overnight with 10 nM IFNγ (Peprotech), followed by 5 h with 200 ng/ml Lipopolysaccharide from *E. coli* strain K12 (Invivogen). Activated or naive BMDM were isolated, washed, and labeled with 1 μM Cell Tracker Deep Red (Thermo Fisher), Green, or Cell Trace Far Red for 10 min at 37 °C. Labeled cells were then co-cultured in black 96-well plates at variable density with 3T3 cells, keeping the total number at $1.6 \times 10^4$ per well. When blocking TNF with neutralizing antibodies, anti-TNFα (Biolegend) was used at 5 μg/ml.

**Antibody staining**. Cells were fixed on ice for 10 min with 1.6% paraformaldehyde. Fixed cells were washed twice with PBS and permeabilized with ice cold 90% methanol. Fixed and permeabilized cells were washed 3× with PBS and blocked for 1 h at room temperature with blocking buffer (3% BSA, 0.3% Triton-X-100 in PBS). Cells were stained with primary antibodies diluted in blocking buffer overnight at 4 °C. Primary stained cells were washed 3× with blocking buffer then incubated with secondary antibodies for 1 h at room temperature, light protected. Secondary stained cells were washed 3× with blocking buffer, twice with PBS, and counterstained with DAPI. To visualize TNF production by BMDM, cells were cultured for 5.5 h with 10 μg/ml Brefeldin A (Sigma–Aldrich) before fixation and staining.

The primary antibodies used were: Monoclonal anti-Cleaved Caspase-8 (clone D5B2) (Cell Signaling Technologies), monoclonal anti-Cleaved Caspase 3 (clone D175) (Cell Signaling Technologies), monoclonal anti-phospho-STING (clone D1C4T) (Cell Signaling Technologies), monoclonal anti-TNFα (clone MP6-XT22) (Biolegend), and monoclonal anti-Ki-67 (clone 16A8) (Biolegend). The secondary antibodies used were (all from Jackson ImmunoResearch): polyclonal goat anti-Rabbit IgG-Alexa 594 (cat: 111-586-003), polyclonal goat anti-Rabbit IgG-Alexa 647 (cat: 111-606-003), and polyclonal goat anti-Rat IgG-Alexa 488 (cat: 112-546-003).

To detect externalized phosphatidylserine by Annexin V and Caspase-8 activity by FAM-FLICA-FMK, we used Annexin V-Alexa 568 (Thermo Fisher) and the Vybrant FAM Caspase-8 (Thermo Fisher) Assay kit according to manufacturer instructors.

**Generation of 3T3 caspase-8 FRET reporter cells**. The Caspase-8 FRET reporter plasmid pECFP-IETD2x-Venus (Addgene)[21] was cloned into a piggyBac destination vector with a Puromycin resistance cassette (pPB-ECFP-IETD2x-Venus-Puro) using Gateway Cloning. 3T3 cells were stably transfected with a piggyBac H2B-iRFP(710) plasmid and pPB-ECFP-IETD2x-Venus-Puro using Lipofectamine 3000. After 48 h, double positive transfectants were selected using 2 μg/ml Puromycin (Thermo Fisher) and Blasticidin (Thermo Fisher). The brightest double positive cells were sorted by FACS (UCLA Broad Stem Cell Flow Cytometry Core) and expanded for experiments. For imaging, cells were cultured on collagen and fibronectin-coated glass-bottom 96-well plates.

**Mathematical modeling**. The model is based on Gillespie's algorithm[64], with interactions (infection) between neighboring cells. Cells in this model are on a 15 × 15 uniform triangular grid (also known as a hexagonal grid). Each cell can be in one of four states: (H) healthy, (I) infected, (D) dead following infection, or (B) dead but uninfected. The simulation is initialized by infecting the central cell on the grid. Each infected cell has a viral load that grows linearly and deterministically while the cell is alive and infected (Supplementary Fig. 4C). Healthy cells can only be infected by live infected cells. The probability of infection scales with the total viral load of the cell's nearest neighbors (Supplementary Fig. 4D). The death rates of individual cells depend on their infection status and on their exposure to TNF, and are directly interpolated from measured values (Supplementary Fig. 4E, F). The model is fully parameterized in independent experiments. Model parameters, including confidence intervals in determining them are presented in

Supplementary Table 1. An annotated version of the code is presented in Supplementary Note 1 of the supplementary materials. Every model-derived figure in this paper can be regenerated by executing the model provided in Supplementary Note 1 or accessing the model from our GitHub repository: https://github.com/wollmanlab/NatComms2021_SpatialStochSIR (Table 1).

**Model reactions**.

$$
\begin{cases}
H \xrightarrow{k_I = \mathbf{VI} \cdot \frac{1}{1 + \frac{1}{NN} \sum \mathbf{VL}}} I, \\
H \xrightarrow{\beta_b(\mathbf{TNF})} B, \\
I \xrightarrow{\beta_i(\mathbf{TNF})} D.
\end{cases}
$$

The viral load on an infected cell grows linearly as and independently of TNF concentration demonstrated during early stage viral infection (Supplementary Fig. 4C, J)

$$\mathbf{VL}(t) = \mathbf{VL}(0) + \mathbf{VGR} \cdot t$$

**Epifluorescence microscopy**. Short-term or single timepoint imaging was performed using either Nikon Plan Apo λ 10x/0.45 or 20x/0.75 objectives with a 0.7x demagnifier and Nikon Eclipse Ti microscope with a Flir, Chameleon3 CMOS camera. Long-term imaging was performed using a Carl Zeiss Plan-Apochromat 10x/0.45 objective with a 0.63x demagnifier and Carl Zeiss Axiovert 200 M microscope also with a Flir, Chameleon3 camera housed inside an incubator for stable environmental control. All imaging was accomplished using custom automated software written using MATLAB and Micro-Manager. For all time-lapse microscopy experiments, the time intervals between image captures were always either 15 or 20 min. Image acquisition software is available on the GitHub repository: https://github.com/wollmanlab/Scope.

**Analysis of time-lapse microscopy data**. Automated image analysis was accomplished using custom software written in MATLAB. Image analysis software is available from the GitHub repository: https://github.com/wollmanlab/SingleCellVirusProcessing.

We segmented individual cell nuclei using a seeded watershed algorithm applied to the nuclear dye (Hoechst 33342). The intensity of the fluorescent viral capsid protein VP26 was used as a proxy of viral abundance. Incorporation of a nucleic acid stain (Sytox green or red), accompanied with nuclear condensation measured as a sharp increase in Hoechst 33342 intensity were used for calling cell death. For every cell, we extracted viral abundance, death signal, and position. Threshold for virus infection signal was set at 99.99% intensity measured in an uninfected sample. Threshold for death signal was set at 95% intensity of an uninfected, untreated, fresh culture where death is expected to be minimal.

For high MOI experiments, we used the Jonker-Volgenant algorithm to track cells over time[65,66]. A tracked cell was declared infected/dead if it had presented intensities above the aforementioned thresholds for at least 4 out of 5 consecutive timepoints, accounting for noise (Fig. 3A, B, Supplementary Fig. 3D, G). We also performed population level measurements by counting the total number of healthy cells as a function of time (Fig. 3C, Supplementary Fig. 3E, F) which yielded similar but more robust measurements (Supplementary Fig. 3G).

For low MOI experiments, we used the Jonker-Volgenant algorithm to track cells over time[65,66]. A tracked cell was declared infected/dead if it had presented intensities above the aforementioned thresholds for at least 4 out of 5 consecutive timepoints, accounting for noise (Fig. 4E, Supplementary Fig. 4H).

Fractions of healthy, infected, dead following infection, and dead without infection are calculated as a fraction of the initially seeded cells. Cells that were initially seeded but disappeared are assumed to be dead since dead cells often detach from the plate or otherwise disintegrate.

To quantify single-cell infection rate as a function of nearest neighbor viral load (Supplementary Fig. 4D) we co-cultured uninfected target cells with a small fraction of infected cells. We imaged the cells every 20 min, which allows us to detect infection rates as long as they are slower than 3 per hour. At every timepoint, we measured the infection state of individual cells, and the total nearest neighbor viral load (defined as the total viral fluorescent signal from cells within a 70 μm radius of the target cell center) of any uninfected target cell. We then determined the fraction of target cells that were infected during the 20 min interval. Overall, we tracked 335,999 possible infection events, 4% of which resulted in productive infection. We then conditioned our data on the nearest neighbor viral load using 25 equally sized bins, each with ~13,200 potential infection events, and repeated the calculation of fractions of successful infections over the 20 min interval. To assess the robustness of our measurement we calculated the standard error of the mean by repeated sampling (100 iterations) of $n = 500$ cells out of every group and measuring the s.d. of the resulting sample fractions. We then fit the corresponding curve to a Hill function (Supplementary Fig 4D) to extract the viral infectivity, the maximal rate of viral infection given a saturating viral load in the infecting cells, and the EC50 of viral infection, the half maximal viral load of the neighboring infecting cells.

**Mouse cornea viral infection and embedding**. Eyes were collected from R26-H2B-mCherry or C57Bl/6 mice and a small section of the corneal epithelium was disrupted using a 0.5 mm rotating burr (Gulden Ophthalmics). Whole eyes were then incubated overnight with $4 \times 10^6$ PFU of HSV-1 mCerulean or tdTomato in prewarmed cornea media (Fluorobrite with 10% FCS, L-glutamine, 10U/ml Penicillin, 10 μg/ml Streptomycin, and 0.025 μg/ml Amphotericin B). Whole eyes were washed $5 \times 5$ min each in 10 ml room temperature PBS. Corneas were then dissected and embedded in a 0.5 ml syringe in a mixture of 1% low melting point agarose and cornea media supplemented with 1:50,000 Sytox Green and, where indicated, 50 ng/ml TNFα. The embedded corneas were then briefly cooled at 4 °C to solidify the agarose. The embedded cornea was then extruded from the syringe into an incubated chamber containing prewarmed cornea media supplemented with Sytox Green and TNFα for imaging.

**Multicolor light-sheet microscopy**. We designed and built a high speed, multicolor light-sheet microscope specialized for large, live samples imaged over several days (Supplementary Fig. 5A). Due to the large dimensions of our sample, an L-SPIM optical configuration was selected. Using the OpenSPIM design as a template[67–69], we modified several hardware components, and completely rewrote the acquisition software and methodology, to satisfy our unique demands.

**Multicolor imaging**. For multicolor illumination, we use an Omicron LightHUB laser combiner with four lasers:

(1) Omicron LuxX 120 mW 405 nm laser.
(2) Omicron LuxX 100 mW 488 nm laser.
(3) Coherent OBIS 80 mW 561 nm OPSL.
(4) Omicron LuxX 100 mW 638 nm laser.

We adapted the infinity-space tube of the OpenSPIM to house an optical filter changer (Sutter Instruments). We further modified the sample positioning arm of the 4D USB Stage (Picard Industries) to physically accommodate the filter wheel.

**Modified sample chamber and objective**. To allow a larger field of view, combined with enhanced resolution, we opted for a $16 \times 0.8$NA Nikon CFI LWD Plan Fluorite Objective. For this, we modified the sample chamber, the sample chamber holder, and the objective holder ring from the original OpenSPIM design (Pieter Fourie Design and Engineering). For illumination, we used the Olympus UMPLFLN10XW objective.

**Beam pivoting**. To create a uniform excitation profile and minimize streaking artifacts, a 3khz resonant scanner (SC10-HF-6dia-20- 3000, EOPC) was positioned along the beam path, perpendicular to the optical table plane. The scanner was run continuously during acquisition with an amplitude of 3 degrees.

**Environmental control**. Environmental control is achieved by positioning the entire microscope inside a Forma Steri-Cycle $CO_2$ Incubator (Thermo Fisher) kept at 37 °C with 5% $CO_2$.

**Stride-and-Strobe acquisition**. To facilitate high-speed volume acquisition, we implemented a new acquisition mode we call Stride-and-Strobe (Supplementary Fig. 5B). During imaging of a single volume, the sample is continuously moved with a velocity of 250 μm/s perpendicularly to the imaging plane. While moving the sample, the camera is continuously imaging at a frame rate of 84 fps (11.85 ms exposure). During that time, the sample makes a stride of 3 μm. At the beginning of every individual frame exposure, a trigger is sent from the camera to the illumination laser. The laser turns on for a pulse of 2 ms. The distance the sample moves during this exposure 0.5 μm. The axial resolution of our microscope is 1.5 μm. Therefore, this imaging strategy allows us to sample the tissue every 3 μm while ensuring that motion-blur would not affect our image quality. Using this strategy, we can image a volume of $782 \times 937 \times 1500$ μm³ in roughly 6 s. To allow this mode of imaging, the Z-axis actuator of the 4D-Stage (Picard Industries), which moves the sample in the direction perpendicular to the imaging plane was replaced by a Hi-Res (0.75 um step) Z-axis actuator (Picard Industries). To image a whole cornea, we first position the cornea in the transverse orientation in relation to the detection objective. We image the half of the cornea closest to the detection objective by tiling 18–25 image stacks with a minimal overlap of 15% of the FOV. We then rotate the cornea 90° to the en face orientation and image the half closest to the excitation objective in a similar manner. When multicolor imaging is applied, each block is imaged sequentially in different colors before continuing to the next block.

**Image acquisition software**. All imaging was accomplished using custom automated software written using MATLAB and Micro-Manager and is available through GitHub repository: https://github.com/wollmanlab/Scope.

**Data processing**. Initial data processing was done using custom software implemented in Bash, ImageJ and Java (Supplementary Fig. 5C). These steps relied

heavily on the BigStitcher[70] ImageJ plugin. During acquisition, completed image stacks are transferred to a dedicated data analysis server. Upon arrival raw image stacks are converted into a multi-resolution HDF5 dataset. At the end of acquisition, the tiles are stitched together and the different acquisition angles (views) and channels are aligned as previously described[71]. Blob-like objects like cells and beads are detected during this initial processing. In the case of time-lapse datasets, drift correction was applied. To allow for fast and efficient data processing, we implemented an automated and parallelized pipeline that incorporates these steps. Software is available through GitHub repository: https://github.com/wollmanlab/bigstitchparallel.

**Data presentation**. Complete LSM datasets are fused using beads based registration[71] and downsampled for presentation. Downsampled image stacks are rendered and presented using ImageJ and the 3DScript plugin[72] using custom ImageJ Macro scripts. For presentation, intensity levels for the bottom 5% are set to 0 (background). To account for bright outliers, intensity is scaled linearly so the top 0.01% (Nuclear channel) and 0.1% (other channels) of the pixels are saturated.

**Data analysis**. Data analysis is achieved using custom software implemented in MATLAB. Completed HDF5 datasets are accessed using the MIB software package version 2.60[73].

**Quantification and statistical analysis**. All data shown in figures are represented as mean ± standard deviation (s.d.) or mean ± standard error of the mean (s.e.m.). All figure legends specify s.d. or s.e.m. Except where noted, all experiments used 3 or 4 replicates, which are factored into the s.d. In all experiments, the approximate number of cells analyzed per condition/well is ≈500.

**Reporting summary**. Further information on research design is available in the Nature Research Reporting Summary linked to this article.

## Data availability
Original images can be obtained from the corresponding author upon reasonable request. Source data are provided with this paper.

## Code availability
All software and code is publicly available at https://github.com/wollmanlab/Scope, https://github.com/wollmanlab/Metadata, https://github.com/wollmanlab/bigstitchparallel. Code for our mathematical model of viral spread that generates all model-derived figures has been provided with this paper and can also be found at our Github repository.

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

## Acknowledgements

We thank Dr. Alexander Hoffmann (UCLA) for providing certain mice strains, Dr. Prashant Desai (Johns Hopkins University) for providing fluorescent viruses, the CLICC Lux Lab (UCLA) for 3D printing of microscope components, and Zachary Hemminger for technical assistance. The work was funded by NIH grant R01EY024960 to R.W.

## Author contributions

Conceptualization, Formal Analysis, Writing, Visualization: J.O.Y., A.O.Y., R.W.; Investigation: J.O.Y., A.O.Y., E.M.; Software: A.O.Y., R.W.; Supervision, Funding Acquisition: R.W.

## Competing interests

The authors declare no competing interests.
