## [Peer Review File · Nature Communications]

Reviewers' comments:

Reviewer #1 (Remarks to the Author):

The authors propose a model where TNF accelerates cell death of virally infected cells to protect against viral spread. According to the hypothesis, TNF predisposes the cell to a 'pro-death primed state' which can accelerate cell death when the cell encounters danger, like a viral infection. While this concept is interesting, there are some aspects of this study that should be reexamined. The main problem is that it seems that the authors would like readers to conclude that apoptosis is the likely pathway that is being "primed", but have not taken into account the role of non-apoptotic cell death pathways. My reviewing philosophy (especially in the days of COVID) is to provide comments that will help the authors to strengthen their conclusions without major additional experiments, if possible. Unfortunately, I do not believe that this would be possible in this case, and cannot therefore recommend further consideration of this work by the journal.

Specific Comments:

1. The authors state that TNF signaling can have multiple outcomes and therefore 'activation of apoptotic signaling can occur without a corresponding increase in cell death'. This statement is misleading and could be substituted with 'TNF signaling can occur without a corresponding increase in cell death'.
2. The text says, "treatment with TNF (100 ng/ml) for 6 h resulted in minimal cell death". However, figure 2A shows that 45% of cells are death. This is hardly "minimal" and the text should be modified to include the actual number accordingly.
3. Because treatment with TNF + actinomycin D results in death of the majority of cells, the authors hypothesize that TNF induces a 'pro-death primed state' in cells that does not necessarily culminate in cell death. This 'primed state' is defined by DISC assembly, while apoptosis is restricted by TNF-mediated pro-survival transcription. However, it is currently accepted that TNF-mediated apoptosis is restricted at two stages: 1) cIAP1/2 ubiquitylation of RIPK1 which restricts the ability of this protein to participate in assembly of the DISC, and 2) the regulation of caspase-8 at the DISC by FLIP. The actinomycin D effects could equally affect translation of both cIAP1/2 and FLIP. Therefore, both mechanisms should be considered.
4. The authors failed to provide sufficient evidence of the 'primed state'. The main evidence of such state was DISC assembly, for which they used a proximity ligation assay (PLA) using casp-8 and TRAF2 antibodies (figure 2B). However, TRAF2 is known for its role in TNF-transcriptional signaling, but not for apoptosis signaling. In fact because of the nature of the DISC (several proteins directing different pathways) the use of PLA is not informative. DISC assembly is an unavoidable consequence of TNF ligation, and consequently its formation does not support the authors assertion.
5. The authors declare that cIAP1/2 are the key proteins in pro-survival signaling. This is wrong. Possibly they are conflating there IAPOs with XIAP, as stated above, these ubiquitin ligases restrict the formation of the DISC.
6. The authors say that while the DISC was assembled, only a small fraction of cells presented externalized phosphatidylserine, but figure S2D shows an abundance of Annexin V + cells. Here is another point where the authors' comments conflict with their evidence.

The opinion of this reviewer is that the 'primed state' is confounded with activation of the apoptosis pathway, either identified at the early or late stages, in a fraction of the cell population.

7. The main hypothesis of this paper is that TNF-induced 'primed state' allows the cells to accelerate their commitment to death under certain contexts. To test this, the authors used fibroblasts treated with TNF and infected with HSV-1. TNF accelerated the cell death rate in infected fibroblasts in a dose dependent manner. Because the % of false positives (cells that die

without being infected) also increases with TNF concentration, the authors state that there is trade-off between the speed to kill infected cells and % of erroneous cell death as a product of TNF signaling. However, the false positive % could only be reflecting the number of cells that would die at a particular TNF concentration, independently of the infection.

The authors did not provide evidence to support that the presumed trade-off is “tunable and regulated” and therefore the text should be modified accordingly.

8. A more accurate demonstration of the cell death mechanism induced by TNF + HSV-1 infection in fibroblasts is needed. The combination of these factors could be triggering a different cell death pathway than TNF alone, which inherently will have different kinetics. An alternative pathway of particular interest in the context of HSV-1 infection of mouse cells is necroptosis (<https://doi.org/10.1038/s41418-018-0172-x>).

9. The text indicates that “TNF-mediated restriction of viral spread depends on apoptosis (Fig S4A-B)”. However, Figure S4A shows that a combination of zVAD/necrostatin-1 allows sustained infection of cells even at 100 ng/ml TNF. Given the necroptotic pathway driven by TNF + zVAD, other methods are necessary to verify if the observed mechanism is apoptosis or otherwise. WB analysis of death protein markers of apoptosis and necroptosis are recommended. If the death pathway triggered by TNF + HSV-1 is not explored, then a recommendation is made to change the text where ‘cell death’ should preferably be used instead of ‘apoptosis’.

10. It is essential that the figures indicate enough experimental details to facilitate data interpretation. Indicate what cells were used, treatments and readouts. For example, the method for “quantification of cell death” should be stated in the figure legend. If the method doesn’t measure cell death directly, then the y-axis legend in the graph should be changed accordingly.

Reviewer #2 (Remarks to the Author):

Apoptosis is a fundamental process underlying the functioning of the immune system. Understanding the cell apoptosis regulation in a mechanistic way, is a key for proper control of a broad spectrum of immune-related diseases. The study addresses the TNF-based regulation of the rate of commitment and to the specificity of apoptosis of various subsets of target cells in HSV-1 infection. It brings a clear and inspiring evidence of the impact of TNF level on the apoptosis kinetics in the context of a cytopathic infection. The authors have found that the increase in the speed of apoptosis comes at the cost of false positive decisions favoring the apoptosis of uninfected cell. In addition, the speed-accuracy tradeoff is a tunable process. The study provides an important set of quantitative data for identifying the single-cell and cell-population mechanisms operating to limit the spreading of the virus infection.

Major concerns:

1. The role of TNF in restricting virus spread for in vitro system is shown in the study. However, its relevance for an in vivo infection control has to be carefully discussed in the context of other antiviral mechanisms such as the type I interferon (IFN-I) system. IFN-I is a fast response system which functions to protect rather than destroy the target cell population.

2. The TNF destruction of macrophages at the early phase of innate antiviral immune response can affect the development of an adaptive immune response. Hence, the statements, like “Rapid apoptosis following viral detection is highly advantageous...” need to be relaxed.

3. The authors wrote “Cellular signaling networks are often regarded as information processing units... However, the degree to which this is only metaphoric or is a faithful representation of biochemical processes is unclear.” All Systems Biology and Computational Cell Biology studies clearly show that every cell processes the receptor received signal into specific reactions of various nature. Hence, the statement somewhat underestimates the already existing perception of cell

functioning.

4. Are the TNF concentrations used for in vitro cultures close to physiological ones?
5. How robust are the experimental results with respect to the variation in MOI? Is the TNF molecular form important (i.e., the secreted vs membrane-bound TNF)?
6. The details of the mathematical model are not given. The grid shown in Figure 4B is hexagonal rather than triangular as stated in Method Details (Mathematical Modelling). The decision rules implemented in the model are not described. The data-based calibration procedure is not presented.
7. There seem to be a systematic mismatch between the model dynamics and the data as one can see from comparing Fig. 4 D and C (FN-infected, FP-dead uninfected curves). Has the goodness of fit been checked using standard statistical criteria?
8. The distribution shown in Fig. 3B is right-skewed. Why is it approximated by an exponential function? Is the use of an exponential to approximate it really justified?
9. The time-delay in the virus replication cycle is large enough with respect to the observation window of the experiment. Why is it considered in the computational model?
10. The standard deviation bars shown in Fig. 4G above TNF = 1 ng/ml are too wide to make a robust statement about the effect of TNF on effective reproductive rate of the virus.
11. Could the authors elaborate more on the mechanism of TNF concentration-dependent regulation of the apoptosis induction rate?

Minor:

Fig. 5G – there is no legend.

Based on the excellent feedback we have now revised many of the claims in the original manuscript and performed additional experiments to address specific points. To facilitate and ease the process of re-reviewing this manuscript we have color-coded responses in this rebuttal letter. We use **blue text to designate responses to critiques only in this letter**. Where necessary, we included changes to the manuscript, which are indented and put in quotation marks. In these sections, **orange text represents modifications to the original text**, whereas **red text is unchanged from the original**. We have embedded all figures and data related to the points in this letter below in addition to the revisions made in the manuscript.

Reviewer #1

Specific Comments:

1. The authors state that TNF signaling can have multiple outcomes and therefore 'activation of apoptotic signaling can occur without a corresponding increase in cell death'. This statement is misleading and could be substituted with 'TNF signaling can occur without a corresponding increase in cell death'.

We thank the reviewer for pointing that this statement as written could cause confusion. We have completely re-written the section of the paper that describes the TNF-mediated 'primed-to-death' cell state and, in the revised section, this sentence has been removed. The revised section can be found in its entirety as part of the response to critique #4 (below).

2. The text says, "treatment with TNF (100 ng/ml) for 6 h resulted in minimal cell death". However, figure 2A shows that 45% of cells are death. This is hardly "minimal" and the text should be modified to include the actual number accordingly.

We agree that this statement is not sufficiently clear. The fraction of cells that die in response to TNF alone is much smaller in comparison to the fraction of cells that die following treatment with TNF and actinomycin D (nearly 100%). We have updated the text to more precisely reflect this point and it reads as follows:

"Consistent with previous results, exposure to a saturating dose of TNF (100ng/ml) for 6 hours killed only 45% of the cells, yet co-treatment with TNF and Actinomycin D, to block transcription, killed nearly all cells in the population (Fig 2A)."

3. Because treatment with TNF + actinomycin D results in death of the majority of cells, the authors hypothesize that TNF induces a 'pro-death primed state' in cells that does not necessarily culminate in cell death. This 'primed state' is defined by DISC assembly, while apoptosis is restricted by TNF-mediated pro-survival transcription. However, it is currently accepted that TNF-mediated apoptosis is restricted at two stages: 1) cIAP1/2 ubiquitylation of RIPK1 which restricts the ability of this protein to participate in assembly of the DISC, and 2) the regulation of caspase-8 at the DISC by FLIP. The actinomycin D effects could equally affect translation of both cIAP1/2 and FLIP. Therefore, both mechanisms should be considered.

Our intent was not to define 'pro-death primed state' using DISC, rather as a state with increased probability for cell death (sensitized). The existence of such a primed state is directly supported by existing as well as new experiments we performed and we revised the text substantially to clarify this point (see full details in answer to point #4). The experiments related to DISC and cIAP are there to provide some mechanistic details into what this state entails molecularly. We completely agree with the reviewer that the interpretation of Actinomycin D experiment should be better worded as we did not explicitly test for a role of FLIP in restricting TNF mediated death. Our data does support the involvement of cIAP1/2 as shown through the use of IAP antagonist/SMAC mimetic LCL-161, which inhibits cIAP1, cIAP2, and XIAP. These data confirm the role for

clAP1/2 restriction of RIPK1 and TNF-mediated cell death. We now added a sentence to specifically clarify this point.

While the use of LCL-161 recapitulates the qualitative results obtained by Actinomycin D, contributions from other mechanisms, such as regulation of caspase-8 by FLIP were not excluded and could be involved in the creation of 'primed to death' cell state.

4. The authors failed to provide sufficient evidence of the 'primed state'. The main evidence of such state was DISC assembly, for which they used a proximity ligation assay (PLA) using casp-8 and TRAF2 antibodies (figure 2B). However, TRAF2 is known for its role in TNF-transcriptional signaling, but not for apoptosis signaling. In fact because of the nature of the DISC (several proteins directing different pathways) the use of PLA is not informative. DISC assembly is an unavoidable consequence of TNF ligation, and consequently its formation does not support the authors' assertion.

We thank the reviewer for bringing this to our attention. Given the signaling function of DISC (independent of apoptosis signaling), pointed out by the reviewer, we have revised this section of this paper to focus on the role of TNF in eliciting a *functionally* primed state. To this end, we present several experiments. First (A), when cells were treated with a very low concentration of TNF (too low to induce any death), exposure to LCL-161 killed ~50% of the cells (B). Importantly, administering LCL-161 to cells *not* treated with TNF did not result in any cell death.

This indicates that TNF sensitizes a majority of the cells to death that is restrained by IAP proteins (clAP1/2 and XIAP as these are the proteins inhibited by LCL-161). In this experiment, cells were treated (or left untreated) with 0.3ng/ml TNF concurrently with 50pg/ml LCL-161. Cell death was quantified by incorporation of Sytox Green at 24h post-treatment.

Second (C), the sensitized cell state is not dependent on a consistent source of TNF in the media. Demonstrating this, administering LCL-161 to TNF-treated cells ~12h after TNF was washed out caused significant cell death (D). In this experiment, cells were treated with 0 or 0.3ng/ml TNF for 24h and then washed and rested overnight. The next morning, the indicated doses of LCL-161 were added to cells and cell death was quantified the next day by incorporation of Sytox Green.

To further confirm that TNF induces an altered cell state characterized by an increased sensitivity to cell death, we performed additional experiments (E). The goal of these experiments were to more clearly investigate the timescale during which cells are sensitized to death after removal of TNF from the media. Cells were treated overnight with 1ng/ml of TNF, and then washed the next day. Some cells were left untreated as controls. We then added 0.25µg/ml Actinomycin D at different time intervals after TNF was washed and quantified death 4 hours later. As controls, some cells were (1) left unprimed and cultured with DMSO as a vehicle, (2) left unprimed and treated with actinomycin D, and (3) treated only with TNF. These experiments revealed that TNF sensitizes cells to death by actinomycin treatment for a period of 12-24h post-wash (F). By 24 hours post-wash, cells are almost returned to control levels of death. This indicates that TNF treatment induces an altered cell state where they are sensitized to cell death by actinomycin D treatment for a transient period lasting about a day.

This section of the manuscript has been completely rewritten and the above figures (along with others) have been added to a new version of Figure 2 (see below for complete figure) to reflect a focus on a *functionally* primed-to-death state that is induced by TNF treatment. This section of the manuscript reads as follows:

“TNF has a well-established role regulating cell death pathways [1]. Therefore, we investigated whether viral defense is related to TNF increasing the cellular propensity to die. TNF simultaneously activates opposing pro-death signaling and a pro-survival transcriptional response [2–6]. Consistent with previous results, exposure to a saturating dose of TNF (100ng/ml) for 6 hours killed only 45% of the cells, yet co-treatment with TNF and Actinomycin D, to block transcription, killed nearly all cells in the population (Fig 2A) [1]. This suggests that TNF activates death pathway signaling in most of the cells, but the execution of death is restrained by cellular production of pro-survival proteins. What biological function could result from the simultaneous activation of antagonistic pathways? We reasoned that the primary function of TNF may not be to kill cells *per se*, but to shift cells into a “primed” state, conferring an increased propensity to die upon exposure to additional death signals. To test this hypothesis, we co-treated cells with a concentration of TNF too low to cause any cell death (0.3ng/ml), and the Inhibitor of Apoptosis Protein (IAP) antagonist LCL-161 [7] IAPs are key restriction points for TNF-mediated cell death [8–11]: cIAP1 and cIAP2 restrict RIPK1 participation in the death inducing signaling complex [12–14], and XIAP restricts Caspase 3 activation[15]. This experiment revealed that although present at too low of a concentration to kill cells alone, TNF sensitizes cells to be killed by LCL-161 (Fig 2B). Importantly, TNF is necessary for LCL-161 to exert any cytotoxic effect as cells not treated with TNF are insensitive to LCL-161. While the use of LCL-161 recapitulates the qualitative results obtained by Actinomycin D, contributions from other mechanisms, such as regulation of caspase-8 by FLIP were not excluded and could be involved in the creation of ‘primed to death’ cell state.

We next asked if the priming effect of TNF depends on ligand exposure or represents an altered cell state that does not rely on continued receptor ligation. To test this, we treated cells for 24 hours with the same low dose of TNF (0.3ng/ml), then washed cells and rested them overnight (~12h) before treating with LCL-161 (Fig 2C). Importantly, treating 3T3 fibroblasts with TNF does not cause cells to produce it [16]. Even after a rest period of 12 hours, TNF-treated cells retained sensitivity to LCL-161, indicating that constant exposure to the ligand is not necessary for sensitization to death (Fig 2D). Consistent with this result, cultures treated with higher doses of TNF exhibited Caspase 8 cleavage and cell death that persisted for 24-36h after the cytokine was washed out (Fig 2E-F). Finally, we quantified how long it takes for TNF-primed cells to lose this sensitized state. Cells were treated for 24h with 1ng/ml TNF and then washed (Fig 2G). Actinomycin D was added to cells at different timepoints post-wash and cell death was quantified after 4 hours. TNF-treated cells retained sensitivity to Actinomycin-mediated death for hours after the ligand was removed (Fig 2H). This demonstrates that the death-sensitized cell state persists for about 24 hours after removal of TNF. Collectively, these data illustrate that TNF transitions cells into a reversible,

ligand-independent “primed-to-death” cell state in which pro-death pathway activity is counteracted by cellular production of pro-survival factors.”

The full new version of figure 2 that accompanies the rewritten text is here:

5. The authors declare that cIAP1/2 are the key proteins in pro-survival signaling. This is wrong. Possibly they are conflating there IAPOs with XIAP, as stated above, these ubiquitin ligases restrict the formation of the DISC.

We thank the reviewer for pointing out our error - we incorrectly stated that cIAP1/2 are involved in pro-survival signaling. This has been removed from the text in our revised manuscript. The roles of cIAP1/2 and XIAP are clarified in the revised text to read:

“IAPs are key restriction points for TNF-mediated cell death [8–11]: cIAP1 and cIAP2 restrict RIPK1 participation in the death inducing signaling complex [12–14], and XIAP restricts Caspase 3 activation[15].”

6. The authors say that while the DISC was assembled, only a small fraction of cells presented externalized phosphatidylserine, but figure S2D shows an abundance of Annexin V + cells. Here is another point where the authors’ comments conflict with their evidence. The opinion of this reviewer is that the ‘primed state’ is confounded with activation of the apoptosis pathway, either identified at the early or late stages, in a fraction of the cell population.

This concern has been addressed by the experiment described in response to point 4 above. In this experiment, the evidence for the primed state is that cells pulsed with TNF and then washed exhibit sensitivity to LCL-161 an Actinomycin D treatment that persists for a period of about a day before returning to baseline levels. We removed any claims that might create the wrong impression that we define primed to death state using DISC assembly.

7. The main hypothesis of this paper is that TNF-induced ‘primed state’ allows the cells to accelerate their commitment to death under certain contexts. To test this, the authors used fibroblasts treated with TNF and infected with HSV-1. TNF accelerated the cell death rate in infected fibroblasts in a dose dependent manner.

Because the % of false positives (cells that die without being infected) also increases with TNF concentration, the authors state that there is trade-off between the speed to kill infected cells and % of erroneous cell death as a product of TNF signaling. However, the false positive % could only be reflecting the number of cells that would die at a particular TNF concentration, independently of the infection.

The authors did not provide evidence to support that the presumed trade-off is “tunable and regulated” and therefore the text should be modified accordingly.

We thank the reviewer for the opportunity to clarify our manuscript. In the context of a viral infection, TNF is produced by immune cells - primarily macrophages and neutrophils. Indeed, in Figure 1 we show that TNF is secreted by activated macrophages which inhibits viral spread. We show further that TNF also affects uninfected cells by transitioning them to a primed-to-death state. We agree, and have shown, that this state occurs regardless of the source of TNF, whether from an activated immune cell or exogenously supplied. We think that the reviewer’s comment disregards the fact that in an adult animal, the main source of TNF is activated immune cells in the context of an immune response. In other words, the effects of TNF should not be considered independently of infection as it is nearly always present in the context of an immune response to infection.

We have shown in the paper that different doses of TNF tune the decision making process of exposed cells (Fig 3F). It’s further been widely demonstrated that TNF production by macrophages scales with the amount of pathogen present [17,18]. We find these two facts sufficient to conclude that the trade-off is tunable and regulated. We have modified a section of the text explaining this point and it reads as follows:

“In the work presented here, we did not vary the inflammatory signal over time. However, in the context of infection, macrophages dynamically adapt to changes in pathogen load by altering the amount of TNF that is produced [17]. Tunability in the decision strategy will allow cells to dynamically respond to such changes, adding another layer of feedback control.”

8. A more accurate demonstration of the cell death mechanism induced by TNF + HSV-1 infection in fibroblasts is needed. The combination of these factors could be triggering a different cell death pathway than TNF alone, which inherently will have different kinetics. An alternative pathway of particular interest in the context of HSV-1 infection of mouse cells is necroptosis (<https://doi.org/10.1038/s41418-018-0172-x>).

We thank the reviewer for correctly suggesting that we should not commit to a specific mode of regulated cell death. Indeed we can not exclude necroptosis as a possible alternative pathway and have changed the text to reflect that. The key takeaway from our paper is that TNF modulates the speed of cell death during viral infection to restrict viral spread. For this conclusion to be valid, the form of cell death is somewhat inconsequential.

One key point that we do want to highlight is that whatever the death pathways may be, the dynamics we observe excludes the possibility of TNF and HSV-1 acting orthogonally to independently induce cell death. If that were the case, the rate of cell death in the presence of both TNF and HSV-1 will be the sum of the rates for TNF treated cells (without infection) and for HSV-1 infected cells (without TNF) which is not the case (see fig above). The dotted line shows the anticipated death rate for additivity, which is calculated by taking the rate of death for infected cells not treated with TNF, and adding the death rate for uninfected cells treated with

variable doses of TNF [19]. We observe the solid line which greatly exceeds the dotted line, demonstrating the synergistic effect of TNF and HSV-1. We are currently enthusiastically exploring the mechanistic basis of this synergy. This figure has been added to the supplementary materials and a section has been added to the discussion to elaborate on this point. It reads as follows:

“Our data argues that infected cells undergo apoptosis, rather than other forms of cell death. Briefly, cells exhibit active Caspases 8 (Fig S2B-C, S3A-C) and 3 (Fig S2A), and externalize phosphatidylserine (Fig S2D). However, we did not explicitly rule out other relevant forms of cell death, such as necroptosis [63], necrosis [64], or pyroptosis [65]. The key takeaway from our paper is that TNF modulates the speed of cell death during viral infection to restrict viral spread. For this conclusion to be valid, the form of cell death is somewhat inconsequential. One key point is that the dynamics we observe excludes the possibility of TNF and HSV-1 acting orthogonally to independently induce cell death. If that were the case, the rate of cell death in the presence of both TNF and HSV-1 will be the sum of the rates [19] for TNF treated cells (without infection) and for HSV-1 infected cells (without TNF) which is not the case (Fig S2J). TNF and HSV-1 have a synergistic, rather than orthogonal effect on the death rate. The exact molecular mechanism underlying this synergy merits further investigation but has intriguing therapeutic potential.”

9. The text indicates that “TNF-mediated restriction of viral spread depends on apoptosis (Fig S4A-B)”. However, Figure S4A shows that a combination of zVAD/necrostatin-1 allows sustained infection of cells even at 100 ng/ml TNF. Given the necroptotic pathway driven by of TNF + zVAD, other methods are necessary to verify if the observed mechanism is apoptosis or otherwise. WB analysis of death protein markers of apoptosis and necroptosis are recommended. If the death pathway triggered by TNF + HSV-1 is not explored, then a recommendation is made to change the text where ‘cell death’ should preferably be used instead of ‘apoptosis’.

The key takeaways from our paper is that TNF modulates the speed of cell death during viral infection to restrict viral spread. The regulation of speed, however, comes at the expense of increased death of uninfected cells. Hence the basic conclusion is that TNF causes more cells to die fast enough that the virus does not have time to produce mature viruses and spread. For this conclusion to be valid, it is somewhat inconsequential what form of cell death occurs. Therefore, we see it as reasonable to take the advice of this reviewer and have changed the words ‘apoptosis’ to ‘cell death’ throughout the paper.

10. It is essential that the figures indicate enough experimental details to facilitate data interpretation. Indicate what cells were used, treatments and readouts. For example, the method for “quantification of cell death” should be stated in the figure legend. If the method doesn’t measure cell death directly, then the y-axis legend in the graph should be changed accordingly.

We have updated all of the figure legends to include more experimental detail as suggested by the reviewer.

Reviewer #2:

Major concerns:

1. The role of TNF in restricting virus spread for in vitro system is shown in the study. However, its relevance for an in vivo infection control has to be carefully discussed in the context of other antiviral mechanisms such as the type I interferon (IFN-I) system. IFN-I is a fast response system which functions to protect rather than destroy the target cell population.

Type I Interferons are, indeed, one of the most important antiviral cytokines. In vivo, however, the coordinated action of many different cytokines with different protective effects is needed to counteract pathogens. The importance of TNF as an antiviral cytokine is highlighted by the fact that patients on TNF-blocking drugs such as adalimumab, infliximab, and etanercept are more susceptible to viral infections[20]. Notably, herpesviruses are of specific concern in these settings. The antiviral effect of TNF has also been demonstrated in animal models of viral infection[21–25], where antiviral effects operate through inhibition of viral replication[26–29] and through immunomodulatory functions[30]. The relevance of TNF for viral protection is further corroborated by the striking array of viral defenses that have evolved to counter its activity[31]. One could argue that IFN-I is perhaps “preferable” to the host because it exerts its antiviral effect without killing cells, but multiple different biologic effects are needed to overcome viral defenses. For example, many viruses, including HSV-1, have also evolved mechanisms to block the effects of type I IFN [32] and some viruses, such as Hepatitis B virus (HBV) contain adaptations that prevent IFN induction[33]. Indeed, for HBV, TNF is very important for viral clearance[34] and this likely depends on its pro-apoptotic effect[35]. To summarize, viruses encode diverse mechanisms to evade the immune response therefore there must be multiple shots on goal to clear pathogens.

We have updated our manuscript discussion to more thoroughly consider the role of TNF and other cytokines in antiviral immunity and our revised text reads as follows:

“The context specific manner by which TNF acts indicates how the regulation of cellular decision strategies has evolved to provide specific physiological benefits. In our *in vitro* system, TNF had a significant antiviral effect, however *in vivo*, it acts together with other cytokines to restrict viral spread. The relative importance of TNF versus other other cytokines likely varies between different types and stages of viral infection and merits further investigation using *in vivo* experimental models.

The importance of TNF as an antiviral cytokine is demonstrated by the array of viral adaptations that have evolved to counter its activity [31]. Further, the antiviral effect of TNF has been demonstrated in various animal models of viral infection [21–25]. Nevertheless, *in vivo*, the antiviral response is executed by the concerted activity of many different cytokines. Type I Interferons are important antiviral cytokines which exert their effect without killing cells. However, some viruses, such as HSV-1, encode strategies to block its effect [32] and other viruses, such as hepatitis B virus, block its induction altogether [33]. These viral adaptations to Interferons underscore the importance of the immune system to counter infection with many different strategies. To summarize, viruses encode diverse mechanisms to evade the immune response therefore there must be multiple shots on goal to clear pathogens.”

2. The TNF destruction of macrophages at the early phase of innate antiviral immune response can affect the development of an adaptive immune response. Hence, the statements, like “Rapid apoptosis following viral detection is highly advantageous...” need to be relaxed.

The reviewer makes a good point that destruction of virus-infected cells can have multiple effects that are not always advantageous for the host. We have relaxed the text to reflect this and it now reads:

“Rapid apoptosis following viral detection is one way to prevent completion of the viral life cycle [36,37] and therefore we hypothesized that the TNF-induced “primed for death” state allows cells to accelerate their commitment to death in certain contexts.”

3. The authors wrote “Cellular signaling networks are often regarded as information processing units... However, the degree to which this is only metaphoric or is a faithful representation of biochemical processes is unclear.” All Systems Biology and Computational Cell Biology studies clearly show that every cell processes the receptor received signal into specific reactions of various nature. Hence, the statement somewhat underestimates the already existing perception of cell functioning.

The reviewer makes a good point - we have revised the text in that section of the discussion and it now reads:

“Cellular signaling networks are often regarded as information processing units [38–42]. Our work shows that cell fate decisions are subject to the same fundamental principle, the speed-accuracy tradeoff, that governs many other complex decision making systems [43] including social insects [44,45], humans [46,47], and artificial neural networks [48]. The applicability of such a universal principle to cellular signaling provides further evidence that these networks are indeed bona-fide information processing systems.”

4. Are the TNF concentrations used for in vitro cultures close to physiological ones?

Quantifying the concentration of cytokines in tissues has some caveats because it relies on grinding the tissue up and then measuring the cytokines present in the fluid using ELISA, effectively averaging local concentration fields over the entire organ. In mice developing sepsis, tissue concentrations of TNF hover around 0.05ng/ml, whereas in the blood and peritoneal fluid, it was about 5ng/ml [49]. This is similar to concentrations of TNF measured in human serum during sepsis [50]. In other contexts, TNF in the serum was quantified at 0.005ng/ml[51]. In the synovial fluid of patients with rheumatoid arthritis TNF concentrations range from 0.005-0.5ng/ml[52,53]. TNF concentrations in the serum of RA patients undergoing adalimumab therapy, TNF ranged from 0.005ng/ml-5ng/ml[54,55]. During inflammatory responses induced by viral infection, TNF has been measured in the serum from 0.05-0.5ng/ml[28,56–59].

In summary, concentrations of TNF in serum, synovial fluid, and organs range from 0.005-5ng/ml. It's likely that when the tissue is kept intact, TNF reaches higher concentrations in the context of local cellular communication, nevertheless, concentrations greater than ~10ng/ml are likely in supra-physiologic range. We have added a section in the discussion section to communicate this to readers and it reads as follows:

“In the context of an infection *in vivo*, spatial effects on cytokine spread are likely to alter some of the effects we observed *in vitro*. The average concentrations of TNF quantified in blood or serum during viral infection are on the lower end of the range we used in our *in vitro* experiments [28,56–59]. However, concentrations of TNF in local cell-cell communications are likely to reach high concentrations as has been inferred for other cytokines *in vivo* [60], yet tissues are much more heterogeneous because only cells in close proximity to cytokine producing cells are subject to cytokine exposure. We speculate that *in vivo*, spatial constraints on TNF spread likely provides a layer of protection from TNF-mediated death of healthy bystander cells. However, in disease states where positive feedback creates widespread TNF secretion [16], the entire tissue could be subject to TNFs cytotoxic effects.”

5. How robust are the experimental results with respect to the variation in MOI? Is the TNF molecular form important (i.e., the secreted vs membrane-bound TNF)?

We have done similar experiments assessing the death rates with MOI 1 and 10, although only MOI 10 are shown in the paper. In the case of MOI 10, the virus will not spread because the initial bolus of virus is sufficient to infect every cell in culture, so cell death rates, but not spread, can be gleaned from experiments. There are several technical caveats associated with using cells infected with an MOI of 1 to evaluate death rates. First, at high TNF concentrations, very few cells get infected because the virus does not spread. Due to this sparsity, the measurements of the death rate are noisier. Second, the time when cells become initially infected can span 72h, so there is more variability in the cells due to asynchronous infection (a cell infected in the first 12h is different from a cell infected after nearly 3 days in culture). Third, because many of the cells get infected later in the experiment, we do not carry out the experiment long enough to quantify the time between

infection and death. Hence, many cells are infected and still alive at the end of the experiment (because they were infected much later in the experiment to begin with). This introduces error into the measurement. Nevertheless, we observe a dose-dependent reduction in the time between infection and death with death rates ranging from $0.085\text{-}0.45\text{h}^{-1}$ depending on the dose of TNF (A). In this experiment, cells were infected with MOI 1. The times of infection and death were calculated by thresholding on the fluorescence intensity of the virus and the fluorescence intensity of a nucleic acid dye (sytox green) and hoechst 33342.

In the case of viral spread, we have also quantified it by spiking in a very low number of initially infected cells (rather than free virus) (B). In this experiment, one cohort of cells were infected with MOI 6 overnight. These cells were then harvested and washed and co-cultured at very low density with uninfected cells exposed to different doses of TNF. The amount of viral spread was quantified at 48hpi by quantifying the fluorescence of tagged VP26. This result demonstrates that TNF dose-dependently reduces viral spread when the initial source of virus are infected cells rather than a bolus of virus in the media.

We have also shown for MOI 1 and 2 that the fraction of initially infected cells is invariant to the TNF concentration, indicating that TNF exposure does not change the cells' initial infectivity (Fig 1C and S1C).

6. The details of the mathematical model are not given. The grid shown in Figure 4B is hexagonal rather than triangular as stated in Method Details (Mathematical Modelling). The decision rules implemented in the model are not described. The data-based calibration procedure is not presented.

The hexagonal lattice and the triangular lattice are synonymous and both names are common in the literature. We have made changes in the text to reflect that that read as follows:

“Cells in this model are on a 15x15 uniform triangular grid (also known as a hexagonal grid).”

We stress that the model is largely insensitive to the choice of underlying geometry, within reason.

We thank the reviewer for highlighting a source of confusion. The model presented is not rule-based, but rather a stochastic spatial SIR model based on Gillespie's algorithm that treats an infection as a stochastic event caused by exposure to virions from neighboring cells. We have added a table that contains all of the model parameters along with confidence intervals and experimental source. We further added an annotated version of the core model code as an appendix.

7. There seem to be a systematic mismatch between the model dynamics and the data as one can see from comparing Fig. 4 D and C (FN-infected, FP-dead uninfected curves). Has the goodness of fit been checked using standard statistical criteria?

We thank the reviewer for the suggestion to clarify the presentation of the model. Figure 4D-E presents simulation and experimental fractions of the initially seeded cells that are alive and healthy, alive and infected, dead after infection, or dead without being infected at 48h post initial inoculation. We corrected our analysis so that cells that were initially seeded but disappeared are now assumed to be dead since dead cells often detach from the plate or otherwise disintegrate. The figure (4C-D) and text were modified to more clearly describe the data and the analysis and is included in the descriptions below.

The model simulations do not involve fitting as all of the parameters are measured independently. We added an accompanying table that summarizes the values and confidence intervals for all model parameters. We have further calculated a coefficient of determination (R^2) between the data and the model results of 0.69. We find it remarkable that a simple model with no free parameters matched independently measured data to such a high degree. The following was added to the main text:

“The model also captures a global manifestation of the speed-accuracy tradeoff: it predicts a sweet spot of TNF concentrations that optimizes the balance of costs and benefits (Fig 4D). This prediction was validated by infecting fibroblasts with a low MOI of HSV-1 in the presence of different doses of TNF and observing a similar sweet spot ($R^2=0.69$, Fig 4E, Fig S4H). We find it remarkable that a simple model with no free parameters matched independently measured data to such a high degree.”

And to the methods section:

“The model is fully parameterized in independent experiments. Model parameters, including confidence intervals in determining them are presented in Sup. Table 1. An annotated version of the code is presented in Appendix 1 of the supplementary materials.”

We have added a more detailed description of the image analysis pipeline of infection and cell death to our methods section:

“Time-lapse epifluorescence microscopy of viral infection and cell death - data analysis

Automated image analysis was accomplished using custom software written in MATLAB. Image analysis software is available from the GitHub repository: <https://github.com/wollmanlab/SingleCellVirusProcessing>. We segmented individual cell nuclei using a seeded watershed algorithm applied to the nuclear dye (Hoechst 33342). The intensity of the fluorescent viral capsid protein VP26 was used as a proxy of viral abundance. Incorporation of a nucleic acid stain (Sytox green or red), accompanied with nuclear condensation measured as a sharp increase in Hoechst 33342 intensity were used for calling cell death. For every cell, we extracted viral abundance, death signal, and position. Threshold for virus infection signal was set at 99.99% intensity measured in an uninfected sample. Threshold for death signal was set at 95% intensity of an uninfected, untreated, fresh culture where death is expected to be minimal.

For high MOI experiments, we used the Jonker-Volgenant algorithm to track cells over time [61,62]. A tracked cell was declared infected/dead if it had presented intensities above the aforementioned thresholds for at least 4 out of 5 consecutive timepoints, accounting for noise (Fig 3A,B, Fig S2D). We also performed population level measurements by counting the total number of healthy cells as a function of time (Fig 3C , Fig S2 E-G) which yielded similar but more robust measurements (Fig S2G).

For low MOI experiments (4E S4H) cell tracking was performed as described above. Fractions of healthy, infected, dead following infection, and dead without infection are calculated as a fraction of the initially seeded cells. Cells that were initially seeded but disappeared are assumed to be dead since dead cells often detach from the plate or otherwise disintegrate.”

8. The distribution shown in Fig. 3B is right-skewed skewed. Why is it approximated by an exponential function? Is the use of an exponential to approximate it really justified?

The single-cell distributions show a clear exponential tail that is easier to see when observed in log scale. These distributions are noisy due to the low numbers of infected cells in low MOI and high TNF conditions, and due to the challenge of tracking dying cells. To parametrize the model we used high MOI conditions where all the cells in a dish are uniformly infected and measure the drop off in live cell numbers. Using the population measurements, the exponential drop off in cell numbers are very clearly evident (Figs S3D-E) and the measurement is much more robust. For clarity, this figure has been added to the supplementary figures (S3D) Figures 3B-C and S3F demonstrate that these are corresponding measurements, as one would also expect from first principles.

9. The time-delay in the virus replication cycle is large enough with respect to the observation window of the experiment. Why is it considered in the computational model?

A time-delay in the virus replication is not explicitly considered in the model and linear viral growth begins immediately after infection. The probability of infection depends on the neighboring cells' viral load and does go up significantly once enough virus has accumulated. This, in effect, creates the delay seen between a cell being infected and a cell being infectious.

10. The standard deviation bars shown in Fig. 4G above TNF = 1 ng/ml are too wide to make a robust statement about the effect of TNF on effective reproductive rate of the virus.

We thank the reviewer for pointing out that the data was not presented clearly. The bars were presented to demonstrate the variability in simulation outcomes due to its stochastic nature. We replaced panel 4G with a violin plot that more clearly shows the distribution of outcomes. We further performed a one-way anova to demonstrate the robustness of the difference between the different groups that yielded a p-value of 0.

11. Could the authors elaborate more on the mechanism of TNF concentration-dependent regulation of the apoptosis induction rate?

We have added text to the discussion section of the manuscript elaborating on the mechanism and it reads as follows:

“Our data argues that infected cells undergo apoptosis, rather than other forms of cell death. Briefly, cells exhibit active Caspases 8 (Fig S2B-C, S3A-C) and 3 (Fig S2A), and externalize phosphatidylserine (Fig S2D). However, we did not explicitly rule out other relevant forms of cell death, such as necroptosis [63], necrosis [64], or pyroptosis [65]. The key takeaway from our paper is that TNF modulates the speed of cell death during viral infection to restrict viral spread. For this conclusion to be valid, the form of cell death is somewhat inconsequential. One key point is that the dynamics we observe excludes the possibility of TNF and HSV-1 acting orthogonally to independently induce cell death. If that were the case, the rate of cell death in the presence of both TNF and HSV-1 will be the sum of the rates [19] for TNF treated cells (without infection) and for HSV-1 infected cells (without TNF) which is not the case (Fig S2J). TNF and HSV-1 have a synergistic, rather than orthogonal effect on the death rate. The exact molecular mechanism underlying this synergy merits further investigation but has intriguing therapeutic potential.”

Minor:

Fig. 5G – there is no legend.

We thank the reviewer for pointing out this oversight and have corrected it.

1. Brenner D, Blaser H, Mak TW. Regulation of tumour necrosis factor signalling: live or let die. *Nat Rev Immunol.* 2015;15: 362–374.
2. Beg AA, Baltimore D. An essential role for NF-kappaB in preventing TNF-alpha-induced cell death. *Science.* 1996;274: 782–784.
3. Hsu H, Huang J, Shu HB, Baichwal V, Goeddel DV. TNF-dependent recruitment of the protein kinase RIP to the TNF receptor-1 signaling complex. *Immunity.* 1996;4: 387–396.
4. Tartaglia LA, Rothe M, Hu YF, Goeddel DV. Tumor necrosis factor's cytotoxic activity is signaled by the p55 TNF receptor. *Cell.* 1993;73: 213–216.
5. Van Antwerp DJ, Martin SJ, Verma IM, Green DR. Inhibition of TNF-induced apoptosis by NF-kappa B. *Trends Cell Biol.* 1998;8: 107–111.
6. Micheau O, Tschopp J. Induction of TNF receptor I-mediated apoptosis via two sequential signaling complexes. *Cell.* 2003;114: 181–190.

7. Weisberg E, Ray A, Barrett R, Nelson E, Christie AL, Porter D, et al. Smac mimetics: implications for enhancement of targeted therapies in leukemia. *Leukemia*. 2010;24: 2100–2109.
8. Aldridge BB, Gaudet S, Lauffenburger DA, Sorger PK. Lyapunov exponents and phase diagrams reveal multi-factorial control over TRAIL-induced apoptosis. *Mol Syst Biol*. 2011;7: 553.
9. Albeck JG, Burke JM, Aldridge BB, Zhang M, Lauffenburger DA, Sorger PK. Quantitative analysis of pathways controlling extrinsic apoptosis in single cells. *Mol Cell*. 2008;30: 11–25.
10. Albeck JG, Burke JM, Spencer SL, Lauffenburger DA, Sorger PK. Modeling a snap-action, variable-delay switch controlling extrinsic cell death. *PLoS Biol*. 2008;6: 2831–2852.
11. Gaudet S, Spencer SL, Chen WW, Sorger PK. Exploring the contextual sensitivity of factors that determine cell-to-cell variability in receptor-mediated apoptosis. *PLoS Comput Biol*. 2012;8: e1002482.
12. Annibaldi A, Wicky John S, Vanden Berghe T, Swatek KN, Ruan J, Liccardi G, et al. Ubiquitin-Mediated Regulation of RIPK1 Kinase Activity Independent of IKK and MK2. *Mol Cell*. 2018;69: 566–580.e5.
13. Bertrand MJM, Milutinovic S, Dickson KM, Ho WC, Boudreault A, Durkin J, et al. cIAP1 and cIAP2 facilitate cancer cell survival by functioning as E3 ligases that promote RIP1 ubiquitination. *Mol Cell*. 2008;30: 689–700.
14. Moulin M, Anderton H, Voss AK, Thomas T, Wong WW-L, Bankovacki A, et al. IAPs limit activation of RIP kinases by TNF receptor 1 during development. *EMBO J*. 2012;31: 1679–1691.
15. Deveraux QL, Takahashi R, Salvesen GS, Reed JC. X-linked IAP is a direct inhibitor of cell-death proteases. *Nature*. 1997;388: 300–304.
16. Bagnall J, Boddington C, England H, Brignall R, Downton P, Alsoufi Z, et al. Quantitative analysis of competitive cytokine signaling predicts tissue thresholds for the propagation of macrophage activation. *Sci Signal*. 2018;11. doi:10.1126/scisignal.aaf3998
17. Gottschalk RA, Martins AJ, Angermann BR, Dutta B, Ng CE, Uderhardt S, et al. Distinct NF- κ B and MAPK Activation Thresholds Uncouple Steady-State Microbe Sensing from Anti-pathogen Inflammatory Responses. *Cell Syst*. 2016;2: 378–390.
18. Paludan SR, Mogensen SC. Virus-cell interactions regulating induction of tumor necrosis factor alpha production in macrophages infected with herpes simplex virus. *J Virol*. 2001;75: 10170–10178.
19. Palmer AC, Sorger PK. Combination Cancer Therapy Can Confer Benefit via Patient-to-Patient Variability without Drug Additivity or Synergy. *Cell*. 2017;171: 1678–1691.e13.
20. Kim SY, Solomon DH. Tumor necrosis factor blockade and the risk of viral infection. *Nat Rev Rheumatol*. 2010;6: 165–174.
21. Sergerie Y, Rivest S, Boivin G. Tumor necrosis factor-alpha and interleukin-1 beta play a critical role in the resistance against lethal herpes simplex virus encephalitis. *J Infect Dis*. 2007;196: 853–860.
22. Ruby J, Bluethmann H, Peschon JJ. Antiviral activity of tumor necrosis factor (TNF) is mediated via p55 and p75 TNF receptors. *J Exp Med*. 1997;186: 1591–1596.
23. Kodukula P, Liu T, Rooijen NV, Jager MJ, Hendricks RL. Macrophage control of herpes simplex virus type 1 replication in the peripheral nervous system. *J Immunol*. 1999;162: 2895–2905.
24. Alejo A, Ruiz-Argüello MB, Pontejo SM, Fernández de Marco MDM, Saraiva M, Hernáez B, et al. Chemokines cooperate with TNF to provide protective anti-viral immunity and to enhance inflammation. *Nat Commun*. 2018;9: 1790.
25. Pavic I, Polic B, Crnkovic I, Lucin P, Jonjic S, Koszinowski UH. Participation of endogenous tumour

- necrosis factor in host resistance to cytomegalovirus infection. *J Gen Virol.* 1993;74: 2215–2223.
26. Feduchi E, Alonso MA, Carrasco L. Human gamma interferon and tumor necrosis factor exert a synergistic blockade on the replication of herpes simplex virus. *J Virol.* 1989;63: 1354–1359.
 27. Chen SH, Oakes JE, Lausch RN. Synergistic anti-herpes effect of TNF-alpha and IFN-gamma in human corneal epithelial cells compared with that in corneal fibroblasts. *Antiviral Res.* 1994;25: 201–213.
 28. Seo SH, Webster RG. Tumor necrosis factor alpha exerts powerful anti-influenza virus effects in lung epithelial cells. *J Virol.* 2002;76: 1071–1076.
 29. Mestan J, Digel W, Mitnacht S, Hillen H, Blohm D, Möller A, et al. Antiviral effects of recombinant tumour necrosis factor in vitro. *Nature.* 1986;323: 816–819.
 30. Shrestha B, Zhang B, Purtha WE, Klein RS, Diamond MS. Tumor necrosis factor alpha protects against lethal West Nile virus infection by promoting trafficking of mononuclear leukocytes into the central nervous system. *J Virol.* 2008;82: 8956–8964.
 31. Rahman MM, McFadden G. Modulation of tumor necrosis factor by microbial pathogens. *PLoS Pathog.* 2006;2: e4.
 32. Melroe GT, DeLuca NA, Knipe DM. Herpes simplex virus 1 has multiple mechanisms for blocking virus-induced interferon production. *J Virol.* 2004;78: 8411–8420.
 33. Park S-H, Rehmann B. Immune responses to HCV and other hepatitis viruses. *Immunity.* 2014;40: 13–24.
 34. Chyuan I-T, Hsu P-N. Tumor necrosis factor: The key to hepatitis B viral clearance. *Cellular & molecular immunology.* 2018. pp. 731–733.
 35. Ebert G, Preston S, Allison C, Cooney J, Toe JG, Stutz MD, et al. Cellular inhibitor of apoptosis proteins prevent clearance of hepatitis B virus. *Proc Natl Acad Sci U S A.* 2015;112: 5797–5802.
 36. Roulston A, Marcellus RC, Branton PE. *Viruses and Apoptosis.* 2003 [cited 17 Feb 2020]. doi:10.1146/annurev.micro.53.1.577
 37. Barber GN. Host defense, viruses and apoptosis. *Cell Death Differ.* 2001;8: 113–126.
 38. Koseska A, Bastiaens PI. Cell signaling as a cognitive process. *EMBO J.* 2017;36: 568–582.
 39. Selimkhanov J, Taylor B, Yao J, Pilko A, Albeck J, Hoffmann A, et al. Systems biology. Accurate information transmission through dynamic biochemical signaling networks. *Science.* 2014;346: 1370–1373.
 40. Cheong R, Rhee A, Wang CJ, Nemenman I, Levchenko A. Information transduction capacity of noisy biochemical signaling networks. *Science.* 2011;334: 354–358.
 41. Habibi I, Cheong R, Lipniacki T, Levchenko A, Emamian ES, Abdi A. Computation and measurement of cell decision making errors using single cell data. *PLoS Comput Biol.* 2017;13: e1005436.
 42. Perkins TJ, Swain PS. Strategies for cellular decision-making. *Mol Syst Biol.* 2009;5: 326.
 43. Chittka L, Skorupski P, Raine NE. Speed-accuracy tradeoffs in animal decision making. *Trends Ecol Evol.* 2009;24: 400–407.
 44. Franks NR, Dornhaus A, Fitzsimmons JP, Stevens M. Speed versus accuracy in collective decision making. *Proc R Soc Lond B Biol Sci.* 2003;270: 2457–2463.
 45. Pratt SC, Sumpter DJT. A tunable algorithm for collective decision-making. *Proc Natl Acad Sci U S A.*

2006;103: 15906–15910.

46. Heitz RP. The speed-accuracy tradeoff: history, physiology, methodology, and behavior. *Front Neurosci*. 2014;8: 150.
47. Wickelgren WA. Speed-accuracy tradeoff and information processing dynamics. *Acta Psychol* . 1977;41: 67–85.
48. Huang J, Rathod V, Sun C, Zhu M, Korattikara A, Fathi A, et al. Speed/accuracy trade-offs for modern convolutional object detectors. *arXiv [cs.CV]*. 2016. Available: <http://arxiv.org/abs/1611.10012>
49. Matalaka KZ, Tutunji MF, Abu-Baker M, Abu Baker Y. Measurement of protein cytokines in tissue extracts by enzyme-linked immunosorbent assays: application to lipopolysaccharide-induced differential milieu of cytokines. *Neuro Endocrinol Lett*. 2005;26: 231–236.
50. Damas P, Reuter A, Gysen P, Demonty J, Lamy M, Franchimont P. Tumor necrosis factor and interleukin-1 serum levels during severe sepsis in humans. *Crit Care Med*. 1989;17: 975–978.
51. Nakai Y, Hamagaki S, Takagi R, Taniguchi A, Kurimoto F. Plasma concentrations of tumor necrosis factor-alpha (TNF-alpha) and soluble TNF receptors in patients with anorexia nervosa. *J Clin Endocrinol Metab*. 1999;84: 1226–1228.
52. Nordahl S, Alstergren P, Kopp S. Tumor necrosis factor-alpha in synovial fluid and plasma from patients with chronic connective tissue disease and its relation to temporomandibular joint pain. *J Oral Maxillofac Surg*. 2000;58: 525–530.
53. Larsson S, Englund M, Struglics A, Lohmander LS. Interleukin-6 and tumor necrosis factor alpha in synovial fluid are associated with progression of radiographic knee osteoarthritis in subjects with previous meniscectomy. *Osteoarthritis Cartilage*. 2015;23: 1906–1914.
54. Berkhout LC, l'Ami MJ, Ruwaard J, Hart MH, Heer PO, Bloem K, et al. Dynamics of circulating TNF during adalimumab treatment using a drug-tolerant TNF assay. *Sci Transl Med*. 2019;11. doi:10.1126/scitranslmed.aat3356
55. Nowlan ML, Drewe E, Bulsara H, Esposito N, Robins RA, Tighe PJ, et al. Systemic cytokine levels and the effects of etanercept in TNF receptor-associated periodic syndrome (TRAPS) involving a C33Y mutation in TNFRSF1A. *Rheumatology* . 2006;45: 31–37.
56. Pierce CA, Preston-Hurlburt P, Dai Y, Aschner CB, Cheshenko N, Galen B, et al. Immune responses to SARS-CoV-2 infection in hospitalized pediatric and adult patients. *Sci Transl Med*. 2020;12. doi:10.1126/scitranslmed.abd5487
57. Chen G, Wu D, Guo W, Cao Y, Huang D, Wang H, et al. Clinical and immunological features of severe and moderate coronavirus disease 2019. *J Clin Invest*. 2020;130: 2620–2629.
58. Hayden FG, Fritz R, Lobo MC, Alvord W, Strober W, Straus SE. Local and systemic cytokine responses during experimental human influenza A virus infection. Relation to symptom formation and host defense. *J Clin Invest*. 1998;101: 643–649.
59. Kittigul L, Temprom W, Sujirarat D, Kittigul C. Determination of tumor necrosis factor-alpha levels in dengue virus infected patients by sensitive biotin-streptavidin enzyme-linked immunosorbent assay. *J Virol Methods*. 2000;90: 51–57.
60. Oyler-Yaniv A, Oyler-Yaniv J, Whitlock BM, Liu Z, Germain RN, Huse M, et al. A Tunable Diffusion-Consumption Mechanism of Cytokine Propagation Enables Plasticity in Cell-to-Cell Communication in the Immune System. *Immunity*. 2017;46: 609–620.
61. Jaqaman K, Loerke D, Mettlen M, Kuwata H, Grinstein S, Schmid SL, et al. Robust single-particle tracking in live-cell time-lapse sequences. *Nat Methods*. 2008;5: 695–702.

62. Jonker R, Volgenant A. A shortest augmenting path algorithm for dense and sparse linear assignment problems. *Computing*. 1987;38: 325–340.
63. Nailwal H, Chan FK-M. Necroptosis in anti-viral inflammation. *Cell Death Differ*. 2019;26: 4–13.
64. Kaiser WJ, Upton JW, Mocarski ES. Viral modulation of programmed necrosis. *Curr Opin Virol*. 2013;3: 296–306.
65. Man SM, Karki R, Kanneganti T-D. Molecular mechanisms and functions of pyroptosis, inflammatory caspases and inflammasomes in infectious diseases. *Immunol Rev*. 2017;277: 61–75.

REVIEWER COMMENTS

Reviewer #1 (Remarks to the Author):

In their manuscript, Oyler-Yaniv et al. investigate the role of TNF on viral spread and cell death. Using cell cultures and state-of-the art LSM imaging of whole corneas, they establish a TNF-dependent 'primed-to-death' cell state that determines the speed and accuracy of cell death (i.e. infected vs bystander cell). This tradeoff hinges on both the inflammatory milieu and infection state of the cell. The authors deploy a pseudo-compartmental SIR model parameterized to their data to reveal a system that is almost completely constrained by their quantitative cell culture data. Although, I did not review the prior submission, the manuscript appears to have undergone significant revision under guidance from reviewer comments. Overall, from an experimental standpoint I'm impressed and satisfied with the depth and breadth of the data, and my comments below are intended to be constructive suggestions for final edits. The authors have developed a significant body of work that advances a conceptual leap in understanding cellular decision making, and it should be published in Nature Communications.

-RECL

Comments:

1. The spatial stochastic SIR model developed by authors is simple yet impressive in recapitulating experimental data and might play a significant role in understanding viral infection progression in tissue, but the model is currently understated. Published models often have some kind of post hoc validation where a model prediction is tested experimentally which can give the impression of 'value'. However, this model is serving more of an explanatory role – revealing that complex emergent properties can result from simple and quantifiable relationships, which is a completely valid approach. Since additional experimental characterization of model predictions are likely outside the scope of the manuscript, it can still be improved with discussion to contextualize the model's value i.e. to provide future directions or suggest uses of the model as a predictive tool. For example, the dynamical responses to a different virus with different infectivity (VI in the model) is likely to have an altered relationship with TNF concentration (the sweet spot in 4D). Similarly, the model may have value in understanding viruses with different growth rates (VGR) or at different MOIs. Emphasis of these or some other discussion points related to the model will bring the paper together.
2. The model files should be provided with a run script to call the model with parameters used to produce figure 4D. It's noted that there are 500 simulations per condition, but it's not clear whether variability in the output is due entirely to stochasticity or whether parameters were selected from within the confidence interval or some other distribution. The file would therefore lessen any lingering questions of computational methods.
3. Is there data relating viral load per cell and TNF concentration, i.e., does viral load grow at the same rate for all TNF doses in Fig S4C? If the model assumes viral load is not dependent on TNF, or the opposite, it should be noted.
4. It would be interesting to see the relationship between viral load and cell death e.g. a plot of viral load at the time of death vs time of death of single cells? Would this plot have a downward slope suggesting correlation between viral load and 'death decision latency'? Would it behave differently for different TNF doses?
5. For clarity, the authors should carefully define the meaning of 'viral infectivity' in terms of the axes of Fig. S4D.
6. The methods should explain why low MOI and high MOI experiments are analysed differently.
7. I suggest to using more direct language to "sweet spot" in representing the cost/benefit balance to cells.

8. The x-axis in figure 4D is very close to figure 4E and it looks like the range of Figure 4E is between 0-1000 instead of the intended range of 0-100.

9. Line 224, S2J is probably meant to be S3J.

10. Fig. 5H figure legend is missing.

Reviewer #2 (Remarks to the Author):

In the revised manuscript the authors have productively addressed the comments on the original submission.

There are few minor items which require attention from the authors, e.g. (1) in Table 1 with the model parameters the confidence intervals for the proliferation rates are missing; (2) the reference to source figure (Figures S3,J) needs to be corrected as it is Figure S3,L.

Reviewer #3 (Remarks to the Author):

In their revised manuscript, Oyler-Yaniv, Oyler-Yaniv, Matz and Wollman present experimental evidence, supported by a mathematical model, that, in response to TNF, cells tune their speed-vs-accuracy trade-off in a cell death decision process triggered by viral infection. In experiments using bone marrow derived macrophages with, or without, a knockout of the TNF gene, they show that TNF is required for the restriction of the spread of HSV-1 infection in a population of mouse fibroblasts, even as TNF does not affect the initial viral infection. Next, they characterize a TNF-induced state of increased sensitivity, or 'priming' to cell death, showing that for around 24 hrs following TNF treatment, cells have increased sensitivity to a SMAC-mimetic or to actinomycin D. Combining experiments and mathematical simulations, they show that increasing concentrations of TNF accelerate cell death, resulting in fewer virus-infected cells albeit at the cost of the loss of some uninfected cells (false positives). Finally, using state-of-the-art time-lapse imaging of 3D samples, they show that this trade-off can also be observed ex vivo, in HSV-1 infection of mouse retina.

Overall, in this revised version, the conclusions reached by the authors are carefully stated and well supported by the data, from a collection of technically challenging and well-executed experiments. The new data visualizations (e.g. Fig S3D), and new data (e.g. in Fig 2) provide additional support and context for these conclusions. The manuscript should be of interest to many, as it presents a new way of quantitatively and conceptually thinking about the role of TNF (potentially in concert with other cytokines) in how tissues limit infections. These ideas are nicely placed in the greater context of what is currently understood in the discussion of the paper.

A few minor comments toward further strengthening the presentation of the work:

1- In the legends of Fig. 2, 3, 4 and S2, S3, S4, please add mentions of which types of cells were used in the experiments.

2- In the legends of each figure/figure panel, please report numbers of cells/experiments. Although the legends state when standard deviation or standard error of the means are used, without an 'n', there is no context to capture how well that variability was characterized.

3- For Fig 4B – adding a similar set of visualizations of the death of uninfected cells or total cell death (here or in the supplements) could provide another clear visualization of the trade-off between speed and accuracy, as the graph in 4D suggests that with increasing TNF concentration, and thus faster cell death of infected cells, there are more 'false positive', dead uninfected cells.

4- In the figure legends or methods, specify the time interval between image captures in time-lapse imaging experiments to give readers a better idea of the time resolution used to collect the data.

5- Please carefully check figure referencing. Line 111, instead of S3G-H, should be S3G, J, and line 117, Fig S3E does not show false positive rates.

We thank the reviewers for taking the time to thoroughly review our paper and for their thoughtful comments and suggestions. To facilitate the process of re-reviewing this manuscript we have color-coded responses in this rebuttal letter. We use blue text to designate our responses only in this letter. Where necessary, we included changes to the manuscript, which are indented and put in quotation marks. In these sections, orange text represents modifications to the original text, whereas red text is unchanged from the original. We have embedded all figures and data related to the points in this letter below in addition to the revisions made in the manuscript. All changes made in the manuscript have been highlighted in yellow. Please note that yellow highlights also include changes to the manuscript necessary to comply with editorial guidelines requested by Nature Communications.

Reviewer #1:

1. The spatial stochastic SIR model developed by authors is simple yet impressive in recapitulating experimental data and might play a significant role in understanding viral infection progression in tissue, but the model is currently understated. Published models often have some kind of post hoc validation where a model prediction is tested experimentally which can give the impression of ‘value’. However, this model is serving more of an explanatory role – revealing that complex emergent properties can result from simple and quantifiable relationships, which is a completely valid approach. Since additional experimental characterization of model predictions are likely outside the scope of the manuscript, it can still be improved with discussion to contextualize the model’s value i.e. to provide future directions or suggest uses of the model as a predictive tool. For example, the dynamical responses to a different virus with different infectivity (VI in the model) is likely to have an altered relationship with TNF concentration (the sweet spot in 4D). Similarly, the model may have value in understanding viruses with different growth rates (VGR) or at different MOIs. Emphasis of these or some other discussion points related to the model will bring the paper together.

We thank reviewer #1 for their remarks and agree that it would be a great use of our model to use it to explore the biology of the system further. In particular, we agree it would be interesting to vary the viral infectivity (VI) rate as this is something that naturally differs based on the virus type or strain, and, as we have all witnessed recently, emerges naturally during viral evolution within a population. To explore this possibility, we performed a parameter scan by running the model (200 simulations) for different VI rates from $0.05 h^{-1}$ to $48 h^{-1}$.

At very low VI rates the maximal cell survival occurs in the absence of any TNF. In this regime the spread of the virus is so slow that even low concentrations of TNF cause more harm than the virus itself. As we increase the rate of VI, the ‘optimal’ TNF concentration - or the concentration that maximizes healthy cells - increases. Concurrently the prominence of this optimal concentration decreases as more damage is caused both by the

faster spreading virus and by TNFs effect on bystander cells. Finally, beyond a certain level of viral infectivity no amount of TNF can rescue the population which gets annihilated.

We have added the following section to our modeling results section where we describe this parameter scan and its implication on the use of TNF or other pro-apoptotic treatments to treat viral infections:

“Finally, we used the model to explore how TNF impacts the infection dynamics during infection by a more- or less-infectious form of the virus. To this end, we ran the model with varied viral infectivity rates ranging from $0.05 h^{-1}$ to $48 h^{-1}$. At very low VI rates, the maximal cell survival occurs in the absence of TNF. In this regime, the spread of the virus is so slow that any amount of bystander damage will cause more harm than the virus itself. As the rate of viral infectivity increases, the optimal TNF concentration also increases. Concurrently the prominence of this optimal concentration decreases as more damage is caused both by the faster spreading virus and by TNFs effect on bystander cells. Finally, beyond a certain level of viral infectivity no amount of TNF can rescue the population which gets annihilated.”

A similar scan of viral growth rates will, in this case, be largely redundant. The probability of infection is given by $P_i = VI * \frac{1}{1+\frac{1000}{v}}$ (Fig S4D) where v is the nearest neighbor viral load which is assumed to grow linearly with time ($v \approx VGR \cdot \Delta t$) post infection (Fig S4C,J). Since $VGR \approx 10h^{-1}$ and $\Delta t \sim 1 - 10h$, typically $v \ll 1000$ and therefore $P_i \approx VI \cdot VGR \cdot \Delta t \cdot \frac{1}{1000}$. This means that for the most part doubling the viral load while halving infectivity will have a neutral effect on the overall infection probability. A parameter scan of VGR is therefore somewhat redundant given the range of reasonable parameters.

We note that the emergence of a sweet-spot of TNF concentrations that maximizes cell health was a novel and post hoc prediction of our model, although this was not properly emphasized in the text. The model parameters were all gathered from separate direct experiments and we were not aware of the emergence of such a sweet spot prior to constructing the model. In fact, we found this prediction initially puzzling as TNF increases the death rate of *both infected and uninfected cells*, yet in certain concentrations it can improve overall cell health. The puzzle is of course solved by realizing that through rapid killing of infected cells TNF protects bystander cells from infection, at the price of an enhanced bystander death rate. To clarify this result, we have added the following to the text:

“The model also captures a global manifestation of the speed-accuracy tradeoff: it predicts an optimal non-zero TNF concentration that optimizes the balance of costs and benefits (Fig 4D). This prediction seems counterintuitive as TNF increases the death rate of both infected and uninfected cells, yet at certain concentrations it maximizes the fraction of healthy cells. By rapid killing of infected cells, TNF protects bystander cells from infection at the acceptable cost of a slightly enhanced bystander death rate. To test this prediction, we infected fibroblasts with a low MOI of HSV-1 in the presence of different doses of TNF and observed such an optimal concentration and a striking agreement between the predicted and the observed results ($R^2=0.69$, Fig 4E, Fig S4H). We find it remarkable that a simple model with no free parameters matched independently measured data to such a high degree.”

2. The model files should be provided with a run script to call the model with parameters used to produce figure 4D. It's noted that there are 500 simulations per condition, but it's not clear whether variability in the output is due entirely to stochasticity or whether parameters were selected from within the confidence interval or some other distribution. The file would therefore lessen any lingering questions of computational methods.

We have now supplied a run script that calls the model with parameters (hard-coded) and produces all of the model-derived figures. This script can also be accessed at our Github repository here: https://github.com/wollmanlab/NatComms2021_SpatialStochSIR. Variability in the output is due entirely to stochasticity, not selection of parameters from within confidence intervals or another distribution. The script we provided is well-annotated and we hope that it will provide clarity and transparency to readers. We have added a sentence to the modeling results section to quell any potential for confusion and it is copied below.

“All model parameters are based on individual cell decisions and were independently calibrated in separate experiments (Fig S4C-F, Sup. Table 1). Variability in the outcome of individual simulations stems from the stochastic nature of the model.”

3. Is there data relating viral load per cell and TNF concentration, i.e., does viral load grow at the same rate for all TNF doses in Fig S4C? If the model assumes viral load is not dependent on TNF, or the opposite, it should be noted.

The model assumes no direct connection between TNF and viral load as our initial visual assessment of the data revealed no strong relationship. We have now performed a systematic, quantitative analysis of the growth rate of viral load for different concentrations of TNF. Using data from our high MOI experiments where 3T3 cells were treated with different doses of TNF and infected with MOI 10 of HSV-1. Viral intensity was quantified as described in the main text (briefly, by thresholding on the intensity of fluorescent-tagged viral VP26). This analysis confirmed our initial observation that viral growth is largely independent of TNF concentration (see inset). We have added the following text and accompanying figure (designated as Supplementary Figure J) to our description of the model parameters in the methods section of the paper:

“The viral load on an infected cell grows linearly as and independently of TNF concentration demonstrated during early stage viral infection (Fig S4C, J).”

4. It would be interesting to see the relationship between viral load and cell death e.g. a plot of viral load at the time of death vs time of death of single cells? Would this plot have a downward slope suggesting correlation between viral load and 'death decision latency'? Would it behave differently for different TNF doses?

Using data from our high MOI experiments where 3T3 cells were treated with different doses of TNF and infected with MOI 10 of HSV-1. Viral intensity and time of death were quantified as described in the main text (briefly, by thresholding on the intensity of fluorescent-tagged viral VP26, and incorporation of sytox green/Hoechst 33342 intensity). We generated single-cell and population average plots of viral load at death vs time of death and these are shown below. We see an initial increase in viral load at death, due to the culture getting progressively more infected, followed by a somewhat random relationship in later times. This is consistent with a model where infected cells die at a certain rate regardless of their viral load. This is also evident from the observed single-rate exponential decays we observed in the population (Fig. S3E).

We further investigated a possible relation between TNF concentration and viral load at time of death. We did not see any dependence in the distribution of viral loads at death with TNF concentration.

5. For clarity, the authors should carefully define the meaning of ‘viral infectivity’ in terms of the axes of Fig. S4D.

In this experiment, the phrase ‘viral infectivity’ means the maximal rate of viral infection given a saturating viral load in the neighboring infecting cells. This reviewer’s comment revealed that we did not provide sufficient detail in how we extracted this parameter from the experimental data. As such, we have added a paragraph to the methods section that explains this process (copied below).

“To quantify single cell infection rate as a function of nearest neighbor viral load (Fig S4D) we co-cultured uninfected target cells with a small fraction of infected cells. We imaged the cells every 20 minutes, which allows us to detect infection rates as long as they are slower than 3 per hour. At every time point, we measured the infection state of individual cells, and the total nearest neighbor viral load (defined as the total viral fluorescent signal from cells within a 70 μ m radius of the target cell center) of any uninfected target cell. We then determined the fraction of target cells that were infected during the 20 minute interval. Overall, we tracked 335999 possible infection events, 4% of which resulted in productive infection. We then conditioned our data on the nearest neighbor viral load using 25 equally sized bins, each with \sim 13200 potential infection events, and repeated the calculation of fractions of successful infections over the 20 minute interval. To assess the robustness of our measurement we calculated the standard error of the mean by repeated sampling (100 iterations) of $n=500$ cells out of every group and measuring the s.d. of the resulting sample fractions. We then fit the corresponding curve to a Hill function (S4D) to extract the viral infectivity, the maximal rate of viral infection given a saturating viral load in the infecting cells, and the EC50 of viral infection, the half maximal viral load of the neighboring infecting cells.”

6. The methods should explain why low MOI and high MOI experiments are analysed differently.

High MOI experiments were analyzed using single cell tracking and population wide survival measurements. Population-wide survival measurements are possible here because viral infection with MOI=10 results in highly synchronized infection of every cell in the culture. The results of these two methods (single cell tracking and population wide survival) were similar (Fig 3B-C, S3D-G). To parameterize the model, we chose to use rates extracted from population measurements as they do not rely on cell tracking, which introduces errors and biases especially when cultures are progressively dying and accumulate debris. For low MOI measurements, we used the same cell tracking algorithm and parameters as for the High MOI case. However, we could not use population wide measurements since the infection with MOI=1 results in highly variable infection of cells

across the population. Some cells are infected within the first few hours, whereas others are not infected until 2-3 days post-infection. We've added a statement to our methods section to clarify this, which is copied below.

“For high MOI experiments, we used the Jonker-Volgenant algorithm to track cells over time^{65,66}. A tracked cell was declared infected/dead if it had presented intensities above the aforementioned thresholds for at least 4 out of 5 consecutive timepoints, accounting for noise (Fig 3A-B, S3D,G). We also performed population level measurements by counting the total number of healthy cells as a function of time (Fig 3C , S3E-F) which yielded similar but more robust measurements (Fig S3G).

For low MOI experiments, we used the Jonker-Volgenant algorithm to track cells over time^{65,66}. A tracked cell was declared infected/dead if it had presented intensities above the aforementioned thresholds for at least 4 out of 5 consecutive timepoints, accounting for noise (4E S4H). Fractions of healthy, infected, dead following infection, and dead without infection are calculated as a fraction of the initially seeded cells. Cells that were initially seeded but disappeared are assumed to be dead since dead cells often detach from the plate or otherwise disintegrate.”

7. I suggest to using more direct language to “sweet spot” in representing the cost/benefit balance to cells.

We agree with the critique that it might be more easily understood to a reader if different language is used to refer to this cost-benefit. We suggest a change from the term “sweet spot” to “optimal TNF concentration.” The optimal TNF concentration refers to the concentration at which the fraction of healthy cells is maximized. We have changed this wording throughout the text where the term “sweet spot” originally appeared.

8. The x-axis in figure 4D is very close to figure 4E and it looks like the range of Figure 4E is between 0-1000 instead of the intended range of 0-100.

We have edited this figure to correct this and the modified figure is shown below.

9. Line 224, S2J is probably meant to be S3J.

Thank you for pointing out this error. We have fixed this and have double checked all of the figure references to make sure they are correct.

10. Fig. 5H figure legend is missing.

We have fixed this oversight and the modified figure is shown below.

Reviewer #2:

1. In Table 1 with the model parameters the confidence intervals for the proliferation rates are missing.

We have added the missing values to Table 1 which has been provided with this paper as Supplementary Table I and is also available along with the full model code at our GitHub repository https://github.com/wollmanlab/NatComms2021_SpatialStochSIR.

2. The reference to source figure (Figures S3,J) needs to be corrected as it is Figure S3,L.

Thank you for pointing out this error. We have fixed this and have also double checked all of the figure references to make sure they are correct.

Reviewer #3:

A few minor comments toward further strengthening the presentation of the work:

1. In the legends of Fig. 2, 3, 4 and S2, S3, S4, please add mentions of which types of cells were used in the experiments.

We have added text in each figure legend to clarify which cell types were used in experiments.

2. In the legends of each figure/figure panel, please report numbers of cells/experiments. Although the legends state when standard deviation or standard error of the means are used, without an 'n', there is no context to capture how well that variability was characterized.

We added a statement describing the general range of cells analyzed per condition/well (n≈500 cells) throughout the experiments described. This text reads as follows:

“In all experiments, the approximate number of cells analyzed per condition/well is 500.” (from Quantification and Statistical Analysis of Methods section)

We also added cell numbers for the specific cases where standard error of the mean was used. The revised text for these cases reads as follows:

“To assess the robustness of our measurement we calculated the standard error of the mean by repeated sampling (100 iterations) of n≈500 cells out of every group and measuring the s.d. of the resulting sample fractions.” (from time lapse microscopy analysis methods section)

“Average viral load per cell quantified from HSV-1 fluorescence after infection of 3T3 cells with MOI 10 HSV-1, linear fit of virus accumulation. $n \approx 400$ cells.” (from figure legend S4C)

“The resulting curve was fit to a Hill function. $n \approx 500$ cells” (from figure legend S4D)

“Average viral load per cell quantified from HSV-1 fluorescence after infection of 3T3 cells with MOI 10 HSV-1 in the presence of different doses of TNF, linear fit of virus accumulation in $n \approx 400$ cells.” (from figure legend S4J)

All experiments, aside from a select few noted cases when using animals, used either 3 or 4 replicates to calculate standard deviation. This text reads as follows:

“Except where noted, all experiments used 3 or 4 replicates, which are factored into the s.d” (from Quantification and Statistical Analysis of Methods section)

3. For Fig 4B – adding a similar set of visualizations of the death of uninfected cells or total cell death (here or in the supplements) could provide another clear visualization of the trade-off between speed and accuracy, as the graph in 4D suggests that with increasing TNF concentration, and thus faster cell death of infected cells, there are more ‘false positive’, dead uninfected cells.

Thank you for the suggestion. We generated this figure and agree that it adds information to the plots in 4B specifically by highlighting the tradeoff. This figure has replaced the original 4B and is shown below.

4. In the figure legends or methods, specify the time interval between image captures in time-lapse imaging experiments to give readers a better idea of the time resolution used to collect the data.

All of our time lapse imaging experiments were done using time intervals of 15 or 20 minutes. We have added a sentence to the methods section (the ‘epifluorescence microscopy’ section) to clarify this and the revised text can be read below.

“All imaging was accomplished using custom automated software written using MATLAB and Micro-Manager. For all time lapse microscopy experiments, the time intervals between image captures were always either 15 or 20 minutes. Image acquisition software is available on the GitHub repository: <https://github.com/wollmanlab/Scope>.”

5. Please carefully check figure referencing. Line 111, instead of S3G-H, should be S3G, J, and line 117, Fig S3E does not show false positive rates.

Thank you for pointing out this error. We have fixed this and have double checked all of the figure references to make sure they are correct.

REVIEWERS' COMMENTS

Reviewer #1 (Remarks to the Author):

Thank you to the authors for their improvements to the manuscript. This is a solid work and revisions have addressed my concerns.